# Decoupling global biases and local interactions between cell biological variables

Assaf Zaritsky[1,2], Uri Obolski[3], Zhuo Gan[1,2], Carlos R Reis[1], Zuzana Kadlecova[1], Yi Du[2], Sandra L Schmid[1], Gaudenz Danuser[1,2]*

[1]Department of Cell Biology, UT Southwestern Medical Center, Dallas, United States; [2]Department of Bioinformatics, UT Southwestern Medical Center, Dallas, United States; [3]Department of Zoology, University of Oxford, Oxford, United Kingdom

**Abstract** Analysis of coupled variables is a core concept of cell biological inference, with co-localization of two molecules as a proxy for protein interaction being a ubiquitous example. However, external effectors may influence the observed co-localization independently from the local interaction of two proteins. Such global bias, although biologically meaningful, is often neglected when interpreting co-localization. Here, we describe DeBias, a computational method to quantify and decouple global bias from local interactions between variables by modeling the observed co-localization as the cumulative contribution of a global and a local component. We showcase four applications of DeBias in different areas of cell biology, and demonstrate that the global bias encapsulates fundamental mechanistic insight into cellular behavior. The DeBias software package is freely accessible online via a web-server at https://debias.biohpc.swmed.edu.

*For correspondence: gaudenz. Danuser@utsouthwestern.edu

**Competing interests:** The authors declare that no competing interests exist.

## Introduction

Interpretation of the relations among coupled variables is a classic problem in many areas of cell biology. One example is the spatiotemporal co-localization of molecules – a critical clue to interactions between molecular components; another example is alignment of molecular structures, such as filamentous networks. However, co-localization or alignment may also occur because the observed components are associated with external effectors. For example, the internal components of a polarized cell are organized along the polarization axis, making it difficult to quantify how much of the observed alignment between two filamentous networks is related to common organizational constraints, and how much of it is indeed caused by direct interaction between filaments. Another example is introduced with protein co-localization, where their intensity distributions may be heavily biased to specific levels regulated by the cell state. The combined effects of *global bias* with *local interactions* are manifest in the joint distribution of the spatially coupled variables. The contribution of global bias to this joint distribution can be recognized from the deviation of the marginal distributions of each of the two variables from an (un-biased) uniform distribution.

Although global bias can significantly mislead the interpretation of co-localization and co-orientation measurements, most studies do not account for this effect (*Adler and Parmryd, 2010*; *Bolte and Cordelières, 2006*; *Costes et al., 2004*; *Das et al., 2015*; *Dunn et al., 2011*; *Kalaidzidis et al., 2015*; *Rizk et al., 2014*; *Serra-Picamal et al., 2012*; *Tambe et al., 2011*). Previous approaches indirectly assessed spatial correlations (e.g., [*Drew et al., 2015*; *Karlon et al., 1999*]), variants of mutual information (e.g., [*Krishnaswamy et al., 2014*; *Reshef et al., 2011*]) or spatial biases (*Helmuth et al., 2010*) but did not explicitly quantify the contribution of the global bias to

**eLife digest** Cell biologists often use microscopes to look closely at cells and see what is happening during an experiment. Cell biology experiments typically involve measuring more than one aspect of the cells, for example, the forces a cell is experiencing and the direction it is moving, or the locations of two different components in the cell. The task is then to decipher the interactions between these independent variables to better understand the inner workings of a living cell.

This task, however, can be challenging because other variables can mask the interactions between the pairs of variables being studied. For example, it is difficult to know if two components of cells overlap in microscopy images because they directly interact or simply because the overall organization of the cell makes it more likely they will end up in the same place. Several statistical methods have been developed to estimate and eliminate such confounding effects to reveal specific interactions. However, these confounding effects – termed "global biases" – may themselves contain valuable information about how cells work. Ignoring the possible roles of global biases may limit our understanding of biological processes.

Zaritsky et al. have now addressed this issue by developing an algorithm called DeBias that takes global bias into account by decoupling it from true interactions between two variables. DeBias uses global bias as a second measurement to understand how two biological variables interact via confounding variables. Zaritsky et al. then used DeBias with data from four different cell biology experiments to show that the algorithm works. For example, animal cells contain several proteins that form filaments, which give them their shape and help them move. So-called vimentin filaments and microtubules had previously been seen to occur in the same place within cells but it was not clear whether their alignment was due to local interactions or determined by the overall shape of the cell. DeBias was used to analyze cells that were either still or moving in a particular direction. The algorithm could tease apart the effect of this movement and showed that co-alignment of vimentin filaments and microtubules was caused more by the movement and shape of the cell and less by specific interactions.

Overall, DeBias is a new mathematical tool for cell biologists that is freely accessible online. Other researchers can now use this tool in future studies to identify local interactions and global biases in a wide range of cell biology experiments and interpret the data in a meaningful way.

the observed joint distribution. These methods approach the global bias as a confounding factor (*VanderWeele and Shpitser, 2013*) that must be eliminated for more accurate assessment of the true local interaction, but ignore the possibility that the global bias contains by-itself valuable mechanistic information to cell behavior.

Here, we present *DeBias* as an algorithm to decouple the global bias (represented by a *global index*) from the bona fide local interaction (represented by a *local index*) in co-localization and co-orientation of two independently-measured spatial variables. The decoupling enables simultaneous investigation of processes that drive global bias and local interactions between spatially-matched variables. Our method is dubbed DeBias because it Decouples the global Bias from local interactions between two variables.

To highlight its capabilities, *DeBias* was applied to data from four different areas in cell biology, ranging in scale from macromolecular to multicellular: (1) alignment of vimentin fibers and microtubules in the context of polarized cells; (2) alignment of cell velocity and traction stress during collective migration; (3) fluorescence resonance energy transfer of Protein Kinase C; and (4) recruitment of transmembrane receptors to clathrin-coated pits during endocytosis. These examples demonstrate the generalization of the method and underline the potential of extracting global bias as an independent functional measurement in the analysis of multiplex biological variables.

## Results

### Similarity of observed co-orientation originating from different mechanisms

The issue of separating contributions from global bias and local interactions is best illustrated with the alignment of two sets of variables that carry orientational information. Examples of co-orientation include the alignment of two filament networks (*Drew et al., 2015*; *Gan et al., 2016*; *Nieuwenhuizen et al., 2015*), or the alignment of cell velocity and traction stress, a phenomenon referred to as *plithotaxis* (*Das et al., 2015*; *Tambe et al., 2011*; *Trepat and Fredberg, 2011*). In these systems, global bias imposes a preferred axis of orientation on the two variables, which is independent of the local interactions between the two variables (*Figure 1A*).

Similar observed alignments may arise from different levels of global bias and local interactions. This is demonstrated by simulation of two independent random variables X and Y, representing orientations (*Figure 1B*, left), from which pairs of samples $x_i$ and $y_i$ are drawn to form an alignment angle $\theta_i$ (*Figure 1B*, middle). Then, a local interaction between the two variables is modeled by co-aligning $\theta_i$ by $\zeta_i$ degrees, resulting in two variables $x_i'$ and $y_i'$ with an observed alignment $\theta_i - \zeta_i$ (*Figure 1B*, right).

We show the joint distribution of X, Y for four simulations (*Figure 1C*) where X and Y are normally distributed with identical means but different standard deviations ($\sigma$), truncated to $[-90°, 90°]$, and different magnitudes of local interactions ($\zeta$). The latter is defined as $\zeta = \alpha\theta$ (*Figure 1B*, $\alpha = 1$ for perfect alignment). Throughout the simulations both $\sigma$ and $\alpha$ are gradually increased (*Figure 1C*, left-to-right), implying that the global bias in the orientational variables is reduced while their local interactions increase. As a result, all simulations display similar observed alignments (mean values, 18.9°−19.5°). *Figure 1D* visualizes 100 samples from each of the two most distinct scenarios: low $\sigma$ and no local interaction ($\sigma = 17°$, $\alpha = 0$) leads to tendency of X and Y to align independently to one direction (left); higher variance together with increased interaction ($\sigma = 40°$, $\alpha = 0.5$) leads to more diverse orientations of X and Y (right), while maintaining similar mean alignment. This simple example highlights the possibility of observing similar alignments arising from different mechanisms. While the described properties are well known and many others have used statistical post-processing to eliminate confounding factors for accurate assessment of local interactions (*Drew et al., 2015*; *Helmuth et al., 2010*; *Karlon et al., 1999*; *Krishnaswamy et al., 2014*; *Reshef et al., 2011*), we aim at directly quantifying the global bias, with the goal of extracting encapsulated information that is fundamental to the biological question.

### DeBias: a method to assess the global and local contribution to observed co-alignment

*DeBias* models the observed marginal distributions X′ and Y′ as the sum of contributions by a common effector, i.e., the global bias, and by local interactions that effect the co-alignment of the two variables in every data point (*Figure 2A*).

In a scenario without any global bias or local interaction between X′ and Y′, the observed alignment would be uniformly distributed (denoted *uniform*). Hence any deviation from the uniform distribution would reflect contributions from both the global bias and the local interactions. To extract the contribution of the global bias we constructed a resampled alignment distribution (denoted *resampled*) from independent samples of the marginal distributions X′ and Y′, which decouples matched pairs ($x_i'$, $y_i'$), and thus excludes their local interactions. The global bias is defined as the dissimilarity between the uniform and resampled distributions and accordingly describes to what extent elements of X′ and Y′ are aligned without local interaction (*Figure 2B*). If a local interaction exists then the distribution of the observed alignment angles will differ from the independently resampled alignment distribution. Hence, the uniform distribution will be less similar to the experimentally observed alignment distribution (denoted *observed*) than to the resampled distribution. Accordingly, the local interaction is defined by the difference of dissimilarity between the observed and uniform distributions and dissimilarity between the resampled and uniform distributions (*Figure 2B*).

The Earth Mover's Distance (EMD) (*Peleg et al., 1989*; *Rubner et al., 2000*) was used to calculate dissimilarities between distributions. The EMD of 1-dimensional distributions is defined as the

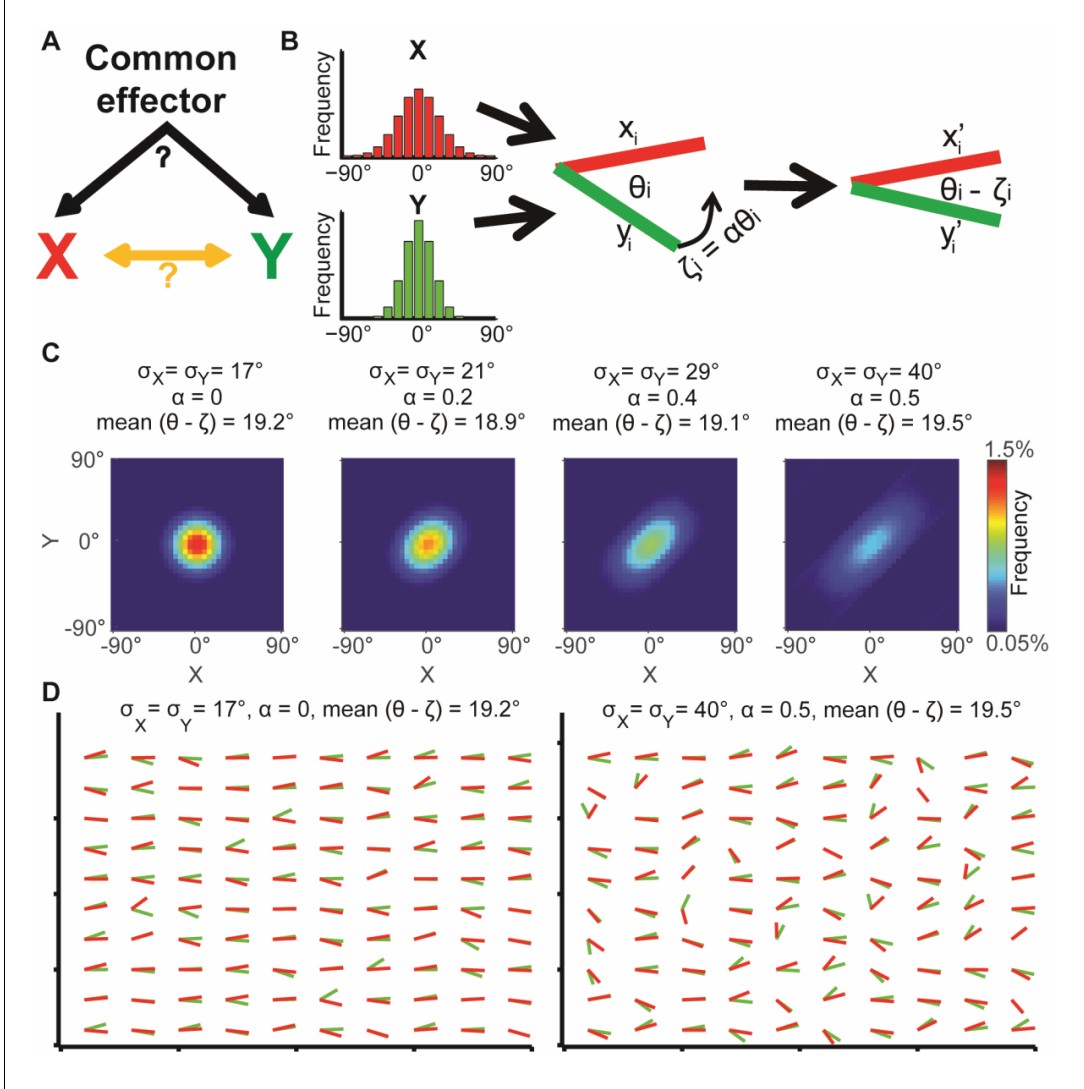

**Figure 1.** Illustration of global bias and local interaction using the alignment of two orientational variables. (A) The relation between two variables X, Y can be explained from a combination of direct interactions (orange) and a common effector. (B) Simulation. Given two distributions X, Y, pairs of coupled variables are constructed by drawing sample pairs $(x_i, y_i)$ and transforming them to $(x_i', y_i')$ by a correction parameter $\zeta_i = \alpha\theta_i$, which represents the effect of a local interaction. $\alpha$ is constant for each of these simulations. (C) Simulated joint distributions. X, Y truncated normal distributions with mean 0 and $\sigma_X = \sigma_Y$. Shown are the joint distributions of 4 simulations with reduced global bias (i.e., increased standard deviation $\sigma_X$, $\sigma_Y$) and increased local interaction (left-to-right). All scenarios have similar observed mean alignment of ~19°. (D) Example of 100 draws of coupled orientational variables from the two most extreme scenarios in panel C. Most orientations are aligned with the x-axis when the global bias is high and no local interaction exists (left), while the orientations are less aligned with the x-axis but maintain the mean alignment between $(x_i', y_i')$ pairs for reduced global bias and increased local interaction (right).

minimal 'cost' to transform one distribution into the other (*Kantorovich and Rubinstein, 1958*). This cost is proportional to the minimal accumulated number of moving observations to adjacent histogram bins needed to complete the transformation. Formally, we calculate $EMD(A, B) = \sum_{i=1,\ldots,K} |\sum_{j=1,\ldots,i} a_j - \sum_{j=1,\ldots,i} b_j|$, with $K$ - number of histogram bins, $a_j$ and $b_j$ - fraction of observations in bin $j$ for distributions $A$ and $B$ correspondingly. Introducing the EMD defines scalar values for the dissimilarities and allows us to define the EMD between resampled and uniform alignment distributions as the *global index* (GI) and the *local index* (LI) as the difference of EMD between observed and uniform and the GI (*Figure 2B*).*Figure 2C*, demonstrates how the GI and LI recognize the global bias and local interactions between the matched variable pairs $(x_i', y_i')$ established in

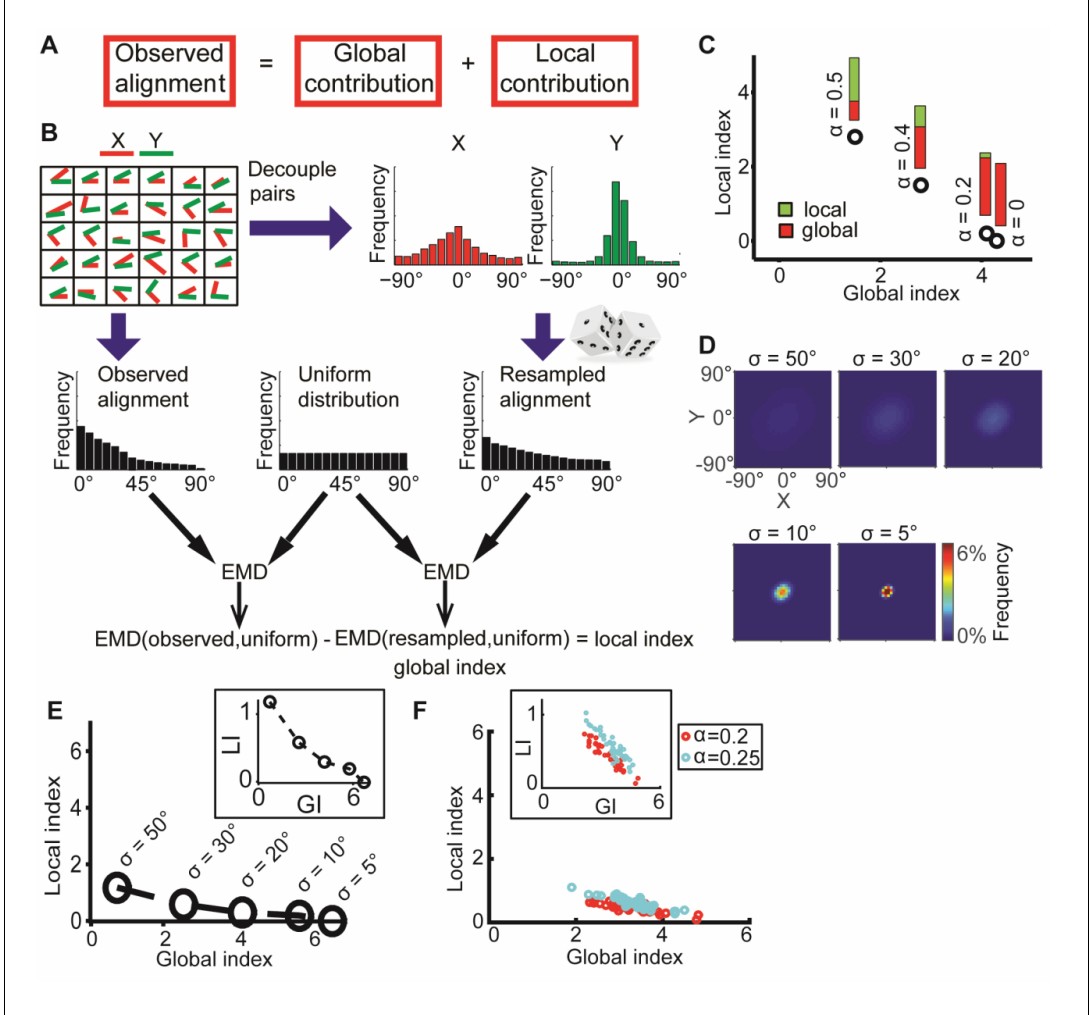

**Figure 2.** DeBias algorithm. (**A**) Underlying assumption: the observed relation between two variables is a cumulative process of a global bias and a local interaction component. (**B**) Quantifying local and global indices: sample from the marginal distributions X, Y to construct the resampled distribution. The global index (GI) is calculated as the Earth Movers Distance (EMD) between the uniform and the resampled distributions. The local index (LI) is calculated as the subtraction of the GI from the EMD between the uniform and the observed distribution. (**C**) Local and global indices calculated for the examples from *Figure 1C*. Black circles represent the (GI,LI) value for the corresponding example in *Figure 1C*, bars represent the relative contribution of the local (green) and global (red) index to the observed alignment. (**D-E**) Simulation using a constant interaction parameter $\alpha = 0.2$ and varying standard deviations of X, Y, $\sigma = 50°$ to 5°. (**D**) Joint distributions. Correlation between X and Y is (subjectively) becoming less obvious for increasing global bias (decreasing $\sigma$). (**E**) GI and LI are negatively correlated: decreased $\sigma$ enhances GI and reduces LI. The change in GI is ~4 fold larger compared to the change in LI indicating that the GI has a limited effect on LI values. Inset: stretched LI emphasizes the negative correlation. (**F**) Both LI and GI are needed to discriminate between simulations with different interaction parameters. $\alpha = 0.2$ (red) or 0.25 (cyan), $\sigma$ is drawn from a normal distribution (mean = 25°, standard deviation = 4°). Number of simulations = 40, for each parameter setting. Inset: stretched LI emphasizes the discrimination. Number of histogram bins, K = 15, for all simulations.

The following figure supplement is available for figure 2:

**Figure supplement 1.** Simulations demonstrating the negative local interactions induce negative local indices.

*Figure 1C*. For a scenario with no local interaction ($\alpha = 0$) DeBias correctly reports LI~0 and GI~3. For a scenario with gradually wider distributions X,Y, i.e., less global bias, and gradually stronger local interactions ($\alpha > 0$), the LI increases while the GI decreases. Additional simulations showed that similar properties apply for negative local interactions $\zeta = \alpha\theta$ (*Figure 1B*) were $\alpha < 0$ (*Figure 2—figure supplement 1*).

In the previous illustrations, changes in spread of the distributions X and Y were compensated by changes in the local interactions. When leaving the interaction parameter α constant while changing the spread of X and Y, a weak, but intrinsically negative correlation between LI and GI becomes apparent (*Figure 2D–E*). Thus, while DeBias can correctly distinguish scenarios with substantial shifts from global bias to local interactions, the precise numerical values estimating the contribution of LI varies between scenarios with a low versus high global bias. To address this issue we propose to exploit the variation between experiments for modeling the relation between LI and GI. This is demonstrated by comparing two distinct values of the interaction parameter, α, emulating different experimental settings (*Figure 2F*). Within experiments variation was obtained by drawing multiple values of σ from a normal distribution. Due to the negative correlation between LI and GI the experimental patterns can only be discriminated by combining LI and GI into a two-dimensional descriptor (*Figure 2F*). This point will be further demonstrated in one of the following case studies and in the Discussion.

## Theory and limiting cases of DeBias

To characterize the properties of DeBias we used theoretical statistical reasoning. The first limiting case is set by the scenario in which observations from X and Y are independent. The expected values of the observed and resampled alignments are identical; accordingly, LI converges to 0 for large N (Appendix 1, Theorem 1). The second limiting case is set by the scenario in which X and Y are both uniform. The corresponding resampled alignment is also uniform; accordingly, GI converges to 0 for a large N (Appendix 1, Theorem 2). The third limiting case occurs with perfect alignment, i.e., $x_i = y_i$ for all i. In this case the observed alignment distribution is concentrated in the bin containing $\theta = 0$. We examine two scenarios of perfect alignment: (1) When all the locally matched measurements are identical ($x_i = y_j$ for all i, j), the resampled distribution is also concentrated in the bin $\theta = 0$ implying that LI = 0 and GI assumes the maximal possible value: $GI = \frac{1}{K} \sum_{i=1,...,K} (i-1) = \frac{K-1}{2}$, where K is the number of quantization bins (Appendix 1, Theorem 3.I). (2) When X, Y are uniform (and $x_i = y_i$ for all i), the resampled distribution is uniform, thus GI = 0 and LI reaches its maximum value: $LI = \frac{1}{K} \sum_{i=1,...,K} (i-1) = \frac{K-1}{2}$, (Appendix 1, Theorem 3.II). Generalizing this case, we prove that LI is a lower bound for the actual contribution of the local interaction to the observed alignment (Appendix 1, Theorem 4). Complementarily, GI is an upper bound for the contribution of the global bias to the observed alignment.

Last, we show that when X and Y are truncated normal distributions, or when the alignment distribution is truncated normal, GI reduces to a limit of 0 as $\sigma \to \infty$, when σ is the standard deviation of the normal distribution before truncation (Appendix 1, Theorem 5). Simulations complement this result demonstrating that σ and GI are negatively associated, i.e., GI decreases with increasing σ (*Figure 2E*). This final property is intuitive, because resampling from more biased distributions (smaller σ) tends to generate high agreement between ($x_i, y_i$) leading to reduced alignment angles and increased GI.

The modeling of the observed alignment as the sum of GI and LI allowed us to assess the performance of DeBias from synthetic data. By using a constant local interaction parameter ζ (ζ = c), we were able to retrieve the portion of the observed alignment that is attributed to the local interaction and to compare it with the true predefined ζ (Appendix 2, *Appendix 2—figure 1*). These simulations demonstrated again the need for a GI-dependent interpretation of LI (first shown in *Figure 2E–F*). Simulations were also performed to assess how the choice of the quantization parameter K (i.e., number of histogram bins) and number of observations N affect GI and LI (Appendix 2, *Appendix 2—figures 2–3*), and this was also verified in our experimental data (*Figure 3—figure supplement 1*). In summary, by combining theoretical considerations and simulations we demonstrated the properties and limiting cases of DeBias in decoupling paired matching variables from orientation data.

## Local alignment of vimentin and microtubule filaments

We applied DeBias to investigate the degree of alignment between vimentin intermediate filaments and microtubules in polarized cells. Recent work using genome-edited Retinal Pigment Epithelial (RPE) cells with endogenous levels of fluorescently tagged vimentin and α-tubulin showed that

vimentin provides a structural template for microtubule growth, and through this maintains cell polarity (*Gan et al., 2016*). The effect was strongest in cells at the wound front where both vimentin

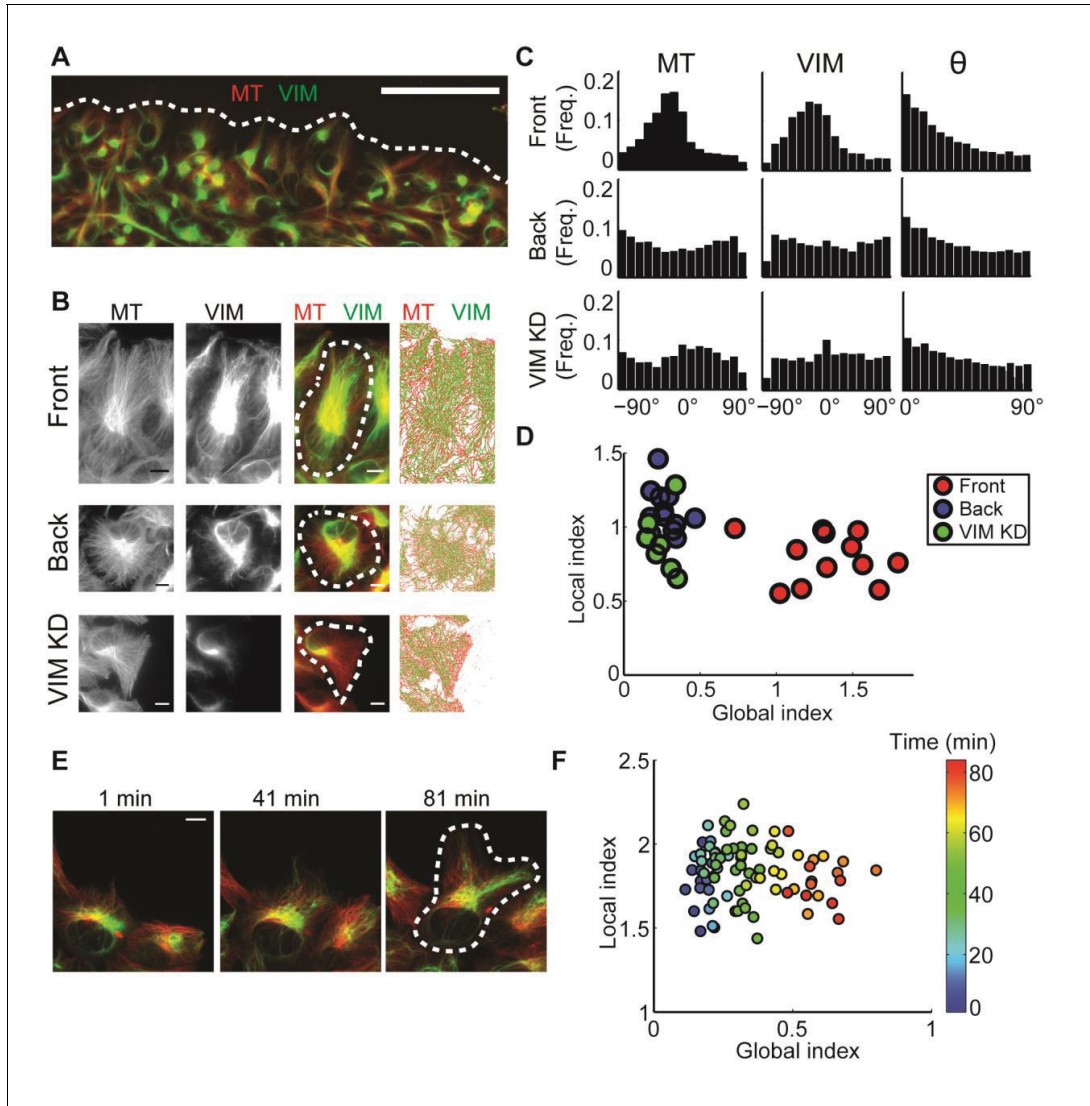

**Figure 3.** : Alignment of microtubule and vimentin intermediate filaments in the context of cell polarity. (**A**) RPE cells expressing TagRFP α-tubulin (MT) and mEmerald-vimentin (VIM) at endogenous levels during a wound healing assay. Scale bar 100 µm. (**B**) Zooming in on cells in different locations in respect to the wound edge. Right-most column, computer segmented filaments of both cytoskeleton systems. Top row, cells located at the wound edge ('Front'); Middle row, cells located 2–3 rows away from the wound edge ('Back'); Bottom row, cells located at the wound edge partially with shRNA knock-down of vimentin. Scale bar 10 µm. (**C**) Orientation distribution of microtubules (left column) and vimentin filaments (middle columns) for the cells outlined in B. Vimentin-microtubule alignment distributions (right column). (**D**) Scatterplot of GI versus LI derived by DeBias. The GI is significantly higher in WT cells at the wound edge ('Front', n = 12) compared to cells inside the monolayer ('Back', n = 12, fold change = 4.8, p-value < 0.0001); or compared to vimentin-depleted cells at the wound edge ('VIM KD', n = 7, fold change = 5.2, p-value < 0.0001). Statistics based on Wilcoxon rank-sum test. All DeBias analyses performed with K = 15. (**E**) Polarization of RPE cells at the wound edge at different time points after scratching. Scale bar 10 µm. (**F**) Representative experiment showing the progression of LI and GI as a function of time after scratching (see color code). Correlation between GI and time ~0.90, p-value < $10^{-30}$ (n time points = 83). N = 5 independent experiments were conducted of which four experiments showed a gradual increase in GI with increased observed polarity. All DeBias analyses performed with K = 15.

The following figure supplement is available for figure 3:

**Figure supplement 1.** LI and GI are independent of the number of observations (N) and the number of histogram bins (K) – experimental evidence from the data in *Figure 3E–F* (time evolution of microtubule-vimentin alignment).

and microtubule networks collaboratively align with the direction of migration (*Figure 3A–C*). An open question remains as to how much of this alignment is caused by the extrinsic directional bias associated with the collective migration of cells into the wound as opposed to a local interaction between the two cytoskeleton systems.

Analysis of the GI and LI revealed that most of the discrepancy in vimentin-microtubule alignment originated from a shift in the global bias (*Figure 3D*), suggesting that the local interaction between the two cytoskeletons is unaffected by the cell position or knock-down of vimentin. Instead, the reduced alignment between the two cytoskeletons is caused by a loss of cell polarity in cells away from the wound edge, probably associated with the reduced geometric constraints imposed by the wound edge. In a similar fashion, reduction of vimentin expression relaxes global cell polarity cues that tend to impose alignment.

To corroborate our conclusion that the global state of cell polarity is encoded by the GI, we performed a live cell imaging experiment, in which single cells at the edge of a freshly inflicted wound in a RPE monolayer were monitored for 80 min after scratching. DeBias was applied to calculate a time sequence of LI and GI. Cells at the wound edge tended to gradually increase their polarity and started migrating during the imaging time frame (*Figure 3E*, *Video 1*). Accordingly, the GI increased over time (*Figure 3F*). We also used this data set to verify that the reported shifts in GI are independent of the number of data points in and the binning of the distribution (*Figure 3—figure supplement 1*). This demonstrates the capacity of DeBias to distinguish fundamentally different effectors of cytoskeleton alignment.

## Identifying molecular factors in alignment of cell velocity and mechanical forces during collective cell migration

Collective cell migration requires intercellular coordination, achieved by mechanical and chemical information transfer between cells. One mechanism for cell-cell communication is plithotaxis, the tendency of individual cells to align their velocity with the maximum principal stress orientation (*He et al., 2015*; *Tambe et al., 2011*; *Zaritsky et al., 2015*). As in the previous example of vimentin and microtubule interaction, much of this alignment is associated with a general directionality of velocity and stress field parallel to the axis of collective migration (*Zaritsky et al., 2015*).

Using a wound healing assay, Das et al. (*Das et al., 2015*) screened 11 tight-junction proteins to identify pathways that promote motion-stress alignment (*Figure 4A*). Knockdown of Merlin, Claudin1, Patj and Angiomotin (Amot) reduced the alignment of velocity direction and stress orientation (*Das et al., 2015*). Further inspection of these hits showed that the stress orientation remained stable upon depletion of these proteins, but the velocity direction distribution was much less biased towards the wound edge (*Zaritsky et al., 2015*). Here, we further analyze this data to demonstrate the capacity of DeBias to pinpoint tight-junction proteins that alter specifically the global or local components that induce velocity-stress alignment.

By distinguishing GI and LI we generated a refined annotation of the functional alteration that depletion of these tight-junction components caused in mechanical coordination of collectively migrating cells (*Figure 4B–C*). First, we confirmed that the four hits reported by (*Das et al., 2015*) massively reduced the GI, consistent with the notion that absence of these proteins diminished the general alignment of velocity to the direction induced by the migrating sheet (*Figure 4B*, red dashed rectangle). Merlin, Patj and Angiomotin reduced the LI to values close to 0, suggesting that the local dependency between stress orientation and

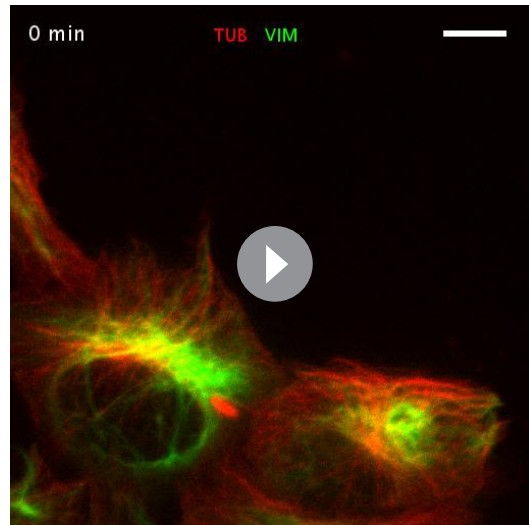

**Video 1.** Polarization of RPE cells at the monolayer edge over time. Please note several occasions (44 and 46 min, 65 and 67 min, 73 and 75 min) of focus drift followed by automated focus correction.

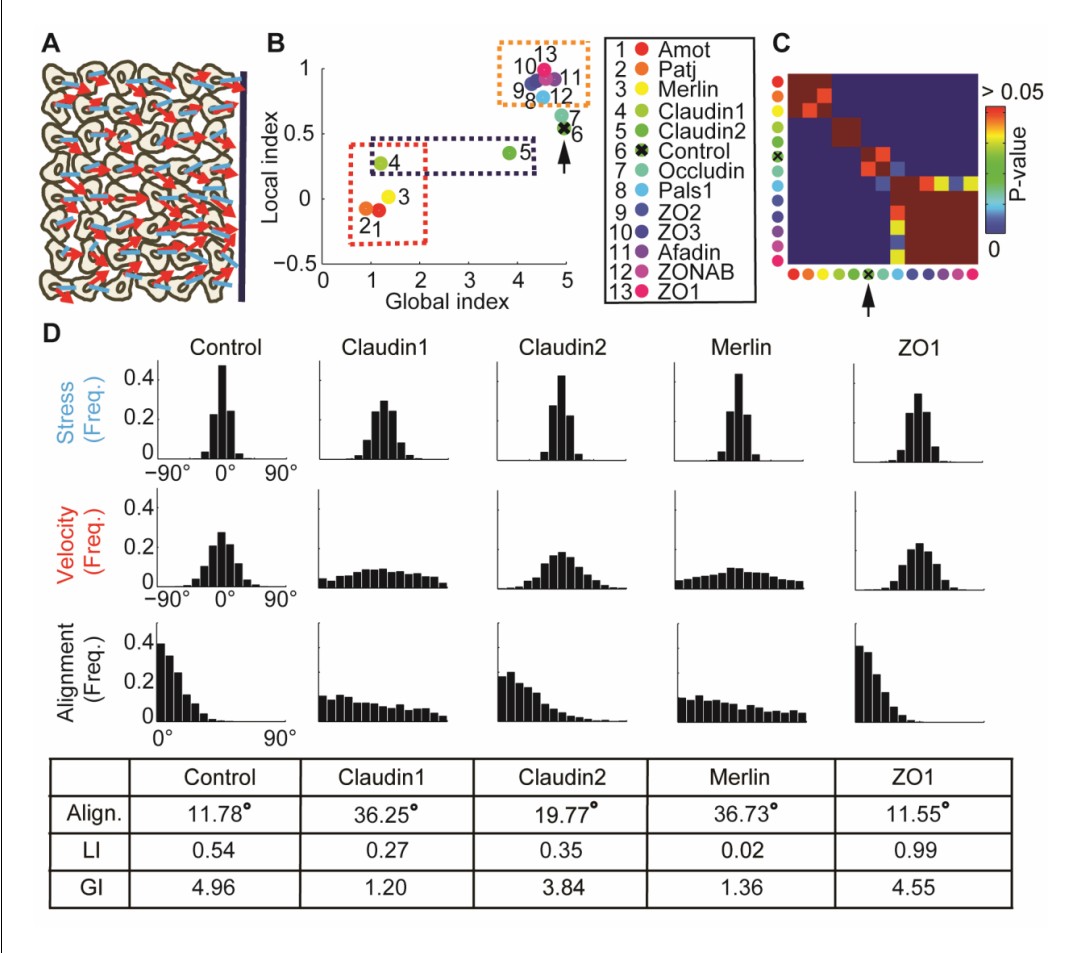

**Figure 4.** : Alignment of stress orientation and velocity direction during collective cell migration. (**A**) Assay illustration. Wound healing assay of MDCK cells. Particle image velocimetry was applied to calculate velocity vectors (red) and monolayer stress microscopy to reconstruct stresses (blue). Alignment of velocity direction and stress orientation was assessed. (**B**) Mini-screen that includes depletion of 11 tight-junction proteins and Merlin. Data from (**Das et al., 2015**), where effective depletion was demonstrated. Shown are GI and LI values; molecular conditions are sorted by the LI values (control is ranked sixth, pointed by the black arrow). Each dot was calculated from accumulation of 3 independent experiments (N = 925–1539 for each condition). Three groups of tight junction proteins are highlighted by dashed rectangles: red - low LI and GI compared to control, purple – different GI but similar LI, orange – high LI. All DeBias analyses were performed with K = 15. (**C**) Pair-wise statistical significance for LI values. P-values were calculated via a permutation-test on the velocity and stress data (Materials and methods). Red – no significant (p ≥ 0.05) change in LI values, blue – highly significant (< 0.01) change in LI values. (**D**) Highlighted hits: Claudin1, Claudin2, Merlin and ZO1. Top: Distribution of stress orientation (top), velocity direction (middle) and motion-stress alignment (bottom). Bottom: table of mean alignment angle, LI and GI. Claudin1 and Claudin2 have similar mechanisms for transforming stress to aligned velocity. ZO1 depletion enhances alignment of velocity by stress.

velocity direction was lost. Depletion of Claudin1, or of its paralog Claudin2, which was not reported as a hit in the Das et al. screen, reduced the LI to a lesser extent, similarly for both proteins, but had very different effects on the GI (*Figure 4B*, purple dashed rectangle). This suggested that the analysis by (*Das et al., 2015*) missed effects that do not alter the general alignment of stress or motion, and implied the existence of a local velocity-stress alignment mechanism that does not immediately change the collective aspect of cell migration.

When assessing the marginal distributions of stress orientation and velocity direction we observed that depletion of Claudin1 reduced the organization of stress orientations and of velocity direction, while Claudin2 reduced only the latter. The LI values of depletion conditions were similar and lower than control (*Figure 4D*). Merlin depletion is characterized by an even lower LI and marginal distributions with aligned stress orientation and almost uniform alignment distribution (*Figure 4D*). Since we

think that aligned stress is transformed to aligned motion (*He et al., 2015*; *Zaritsky et al., 2015*), we propose that in this data the LI quantifies the effect of local mechanical communication on parallelizing the velocity among neighboring cells. Accordingly, stress-motion transmission mechanism is impaired to a similar extent by reduction of Claudin1 and Claudin2, albeit less than by reduction of Merlin.

Using LI as a discriminative measure also allowed us to identify a group of new hits (*Figure 4C*). ZO1, ZO2, ZO3, Occludin and ZONAB are all characterized by small reductions in GI but a substantial increase in LI relative to control (*Figure 4B*, orange dashed rectangle). A quantitative comparison of control and ZO1 depleted cells provides a good example for the type of information DeBias can extract: both conditions yield similar observed alignment distributions with nearly identical means, yet ZO1 depletion has an 83% increase in LI and 8% reduction in GI, i.e., the mild loss in the marginal alignment of velocity or stress is compensated by enhanced local alignment in ZO1 depleted cells (*Figure 4D*). This might point to a mechanism, in which stress orientation is reduced by tight-junction depletion, but enhanced by transmission of stress orientation into motion orientation, leading to comparable alignment. Notably, all paralogs, ZO1, ZO2 and ZO3 fall into the same cluster of elevated LI and slightly reduced GI relative to control experiments. This phenotype is in agreement with the outcomes of a screen that found ZO1 depletion to increase both motility and cell-junctional forces (*Bazellières et al., 2015*).

## Using DeBias to assess protein-protein co-localization

Protein-protein co-localization is another ubiquitous example of correlating spatially matched variables in cell biology. To quantify GI and LI for protein-protein co-localization, we normalized each channel to intensity values between 0 and 1. The 'alignment' $\theta_i$ of matched observations $(x_i,y_i)$ was replaced by the difference in normalized fluorescent intensities $x_i - y_i$ (Materials and methods). Simulations demonstrated that stronger interactions in co-localization are translated to larger LI values and validated that the choice of K (number of histogram bins) and N (number of observations) marginally affect GI and LI (Appendix 3, *Appendix 3—figure 1*). While LI could serve as a measure to assess co-localization, the interpretation of GI is less intuitive. In the following, we present two examples of applying DeBias for protein-protein co-localization, and demonstrate the type of information that can be extracted from the combined GI and LI analysis.

## PKC FRET: a simple example of pixel-based protein-protein co-localization

To test the potential of DeBias in quantification of pixel-based co-localization, we analyzed the effect of fluorescence resonance energy transfer (FRET) in the C kinase activity reporter (CKAR), which reversibly responds to PKC activation and deactivation (*Violin et al., 2003*). Reduced PKC activity leads to energy transfer from CFP to $YFP_{CFP}$, resulting in reduced FRET ratio ($\frac{CFP}{YFP_{CFP}}$) (*Figure 5A*). Assuming that the CFP signal is dominant (CFP > $YFP_{CFP}$), this alteration should reduce the difference between the CFP and $YFP_{CFP}$ channels, which would in DeBias yield an increased LI (*Figure 5A*, Materials and methods).

To test this we labeled hTERT-RPE-1 cells with CKAR and imaged CFP and $YFP_{CFP}$ channels before and after specific inhibition of PKC with HA-100 dihydrochloride (*Figure 5B*, Materials and methods), leading to reduced pixel differences in their normalized fluorescent intensities (*Figure 5C*). As expected, the $\frac{CFP}{YFP_{CFP}}$ ratio decreased (*Figure 5D*), LI values increased (*Figure 5E*) and seemed more sensitive to the FRET. Surprisingly, DeBias indicated a shift in the GI values (*Figure 5F*), reflected in a more homogeneous marginal distribution of both channels before inhibition (*Figure 5G*). Control experiments with cytoplasmic GFP and mCherry expression did not show the shifts observed in LI or GI (*Figure 5H*). Thus, we conclude that PKC inhibition changes the localization of PKC towards a more random spatial distribution. One possible mechanism for this behavior is that deactivation releases the kinase from the substrate. This example illustrates DeBias' capabilities to simultaneously quantify changes in local interaction and global bias in pixel-based co-localization.

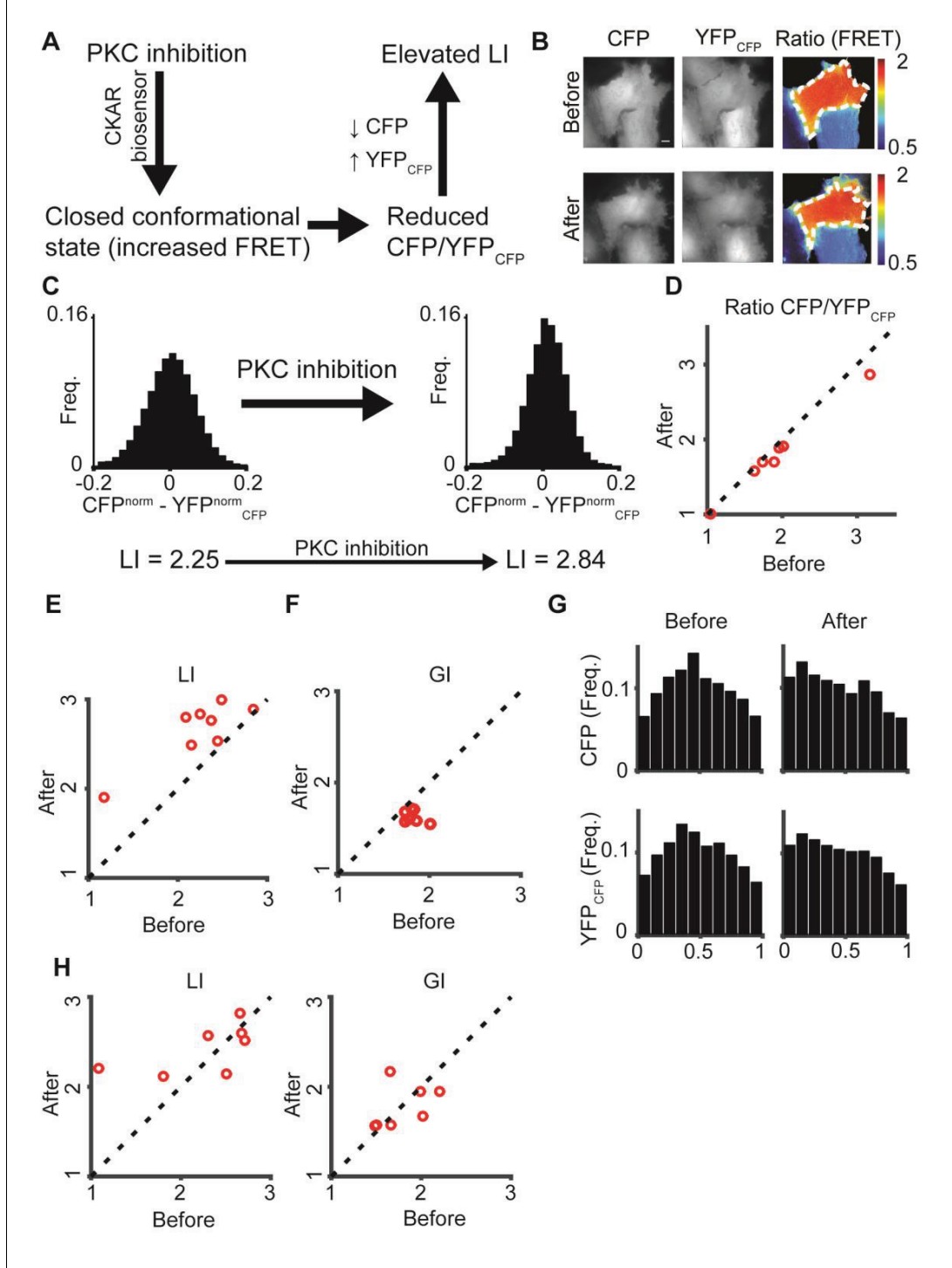

**Figure 5.** : PKC inhibition alters LI and GI. (**A**) PKC inhibition is expected to lead to elevated LI for cells with dominant CFP signal (CFP > $YFP_{CFP}$). Upon FRET, CFP signal is locally transferred to $YFP_{CFP}$, reducing the difference in normalized intensity between the two channels, which increases LI. (**B**) hTERT-RPE-1 cells imaged with the CKAR reporter. A cell before (top) and after (bottom) PKC inhibition. Region of interest was manually annotated and the ratio $\frac{CFP}{YFP_{CFP}}$ was calculated within it. (**C**) Pixel distribution of differences in normalized fluorescent intensities $CFP^{norm}$ - $YFP^{norm}_{CFP}$ before and after PKC inhibition for the cell from panel B. PKC inhibition shifted the average absolute difference from 0.054 to 0.042 and the LI from 2.25 to 2.84. (**D–F**) PKC inhibition experiment. N = 8 cells. Statistics based on Wilcoxon sign-rank test. (**D**) The FRET ratio $\frac{CFP}{YFP_{CFP}}$ decreased (p-value < 0.008), (**E**) LI increased (p-value < 0.008), and (**F**) GI decreased (p-value < 0.008) after PKC inhibition. (**G**) Marginal distribution of CFP and $YFP_{CFP}$ before (top) and

*Figure 5 continued on next page*

Figure 5 continued

after (bottom) PKC inhibition. (**H**) Control experiment. N = 7 cells. hTERT-RPE-1 cells expressing cytoplasmic GFP and mCherry before and after PKC inhibition. No significant change in LI or GI was observed. All DeBias analyses were performed with K = 19.

## Inferring co-localization of molecular cargo and clathrin-coated pits during endocytosis

Clathrin-mediated endocytosis (CME) is the major pathway for entry of cargo receptors into eukaryotic cells. Cargo receptor composition plays an important role in regulating clathrin-coated pit (CCP) initiation and maturation (*Liu et al., 2010*; *Loerke et al., 2009*). The clustering of transferrin receptors (TfnR), the classic cargo receptor used to study CME, promotes CCP initiation, in concert with clathrin and adaptor proteins (*Liu et al., 2010*). Recent evidence suggests a diversity of mechanisms regulating endocytic trafficking, including cross-talks between signaling receptors and components of the endocytic machinery (*Di Fiore and von Zastrow, 2014*). For example, the oncogenic protein kinase Akt has been shown to play an important role in mediating CME in cancer cells (*Liberali et al., 2014*; *Reis et al., 2015*), but not in normal epithelial cells (*Reis et al., 2015*). Here we tested how the decoupling by DeBias of global and local contributions to the overall intensity alignment of clathrin and TfnR, can be used to simultaneously investigate co-localization and predict CCP dynamics, using fixed cell fluorescence imaging.

We used fluorescence images of fixed non-small lung cancer cells (H1299) or untransformed human retinal pigment epithelial cells (ARPE-19) expressing clathrin light chain A fused to eGFP (eGFP-CLCa) as a CCP marker (*Figure 6A–B*). Cells were either treated with DMSO or with an AKT inhibitor (Akt inhibitor X, 'ten'), and imaged by Total Internal Reflection Fluorescence Microscopy (TIRFM). CCPs were reported in the eGFP-CLCa channel and TfnR was visualized by immunofluorescence in a second channel (Materials and methods). For single cells, the location of fluorescent signals of CLCa and TfnR were recorded and the data were pooled and processed by DeBias (Materials and methods).

LI values, indicative of the co-localization between TfnR and CLCa, were significantly lower in Akt-inhibited H1299 compared to control cells (*Figure 6C*). In contrast, Akt inhibition increased the LI values in ARPE-19 cells but this effect was less prominent (*Figure 6D*). Akt inhibition resulted in increased GI values for both cell lines, to a much greater degree in H1299 cells (*Figure 6C–D*). To test whether GI enhances the ability to distinguish between control and Akt-inhibited cells, we applied Linear Discriminative Analysis (LDA) classification to calculate the true positive rate versus the false positive rate for LI alone (black lines, *Figure 6E–F*) or the pair (GI, LI), (orange lines, *Figure 6E–F*). The area under these curves (AUC) provided a direct measure of the ability of each method to accurately classify the experimental condition of single cells. AUC for the (GI, LI) representation was superior to using LI alone for both cell lines (H1299: 0.96 versus 0.88, *Figure 6E*; ARPE-19: 0.83 versus 0.72, *Figure 6F*). Similar benefit in discrimination was achieved when using the GI to complement Pearson's correlation as an alternative to LI for measuring local interaction (*Figure 6—figure supplement 1A–F*). Such improved discrimination is indicative of distinct molecular processes that were altered upon Akt inhibition. We also used this data set to experimentally validate the independence of GI and LI of the number of observations (N, *Figure 6—figure supplement 1G,H*) and the choice of the number of histogram bins (K, *Figure 6—figure supplement 1I,J*).

To interpret the increased GI values for Akt-inhibited cells, we examined the joint and marginal distributions of CLCa and TfnR. Upon Akt-inhibition, the joint distributions were more biased toward regions of low TfnR intensities (*Figure 6G–H*). This was clearly observed in the marginal distributions (*Figure 6I–J*). Hence, although the CLCa distribution appeared not to change upon AKT inhibition, the frequency of CCPs with fewer TfnRs increased. Given the positive relation between TfnR cargo quantities and CCPs maturation (*Loerke et al., 2009*), we wondered whether the increased frequencies of CCPs containing less TfnR might alter CCPs dynamics. Indeed, live-imaging of H1299 cells showed a higher frequency of CCPs with shorter lifetimes upon Akt-inhibition, which was not seen in normal ARPE-19 cells (*Figure 6K–L*, Materials and methods). It has previously been shown that Akt inhibition reduces the rate of TfnR CME uptake in H1299 cells, but not in ARPE-19 cells (*Reis et al., 2015*], see also *Figure 6M–N*); therefore, these findings indicate that the reduced levels of TfnR in

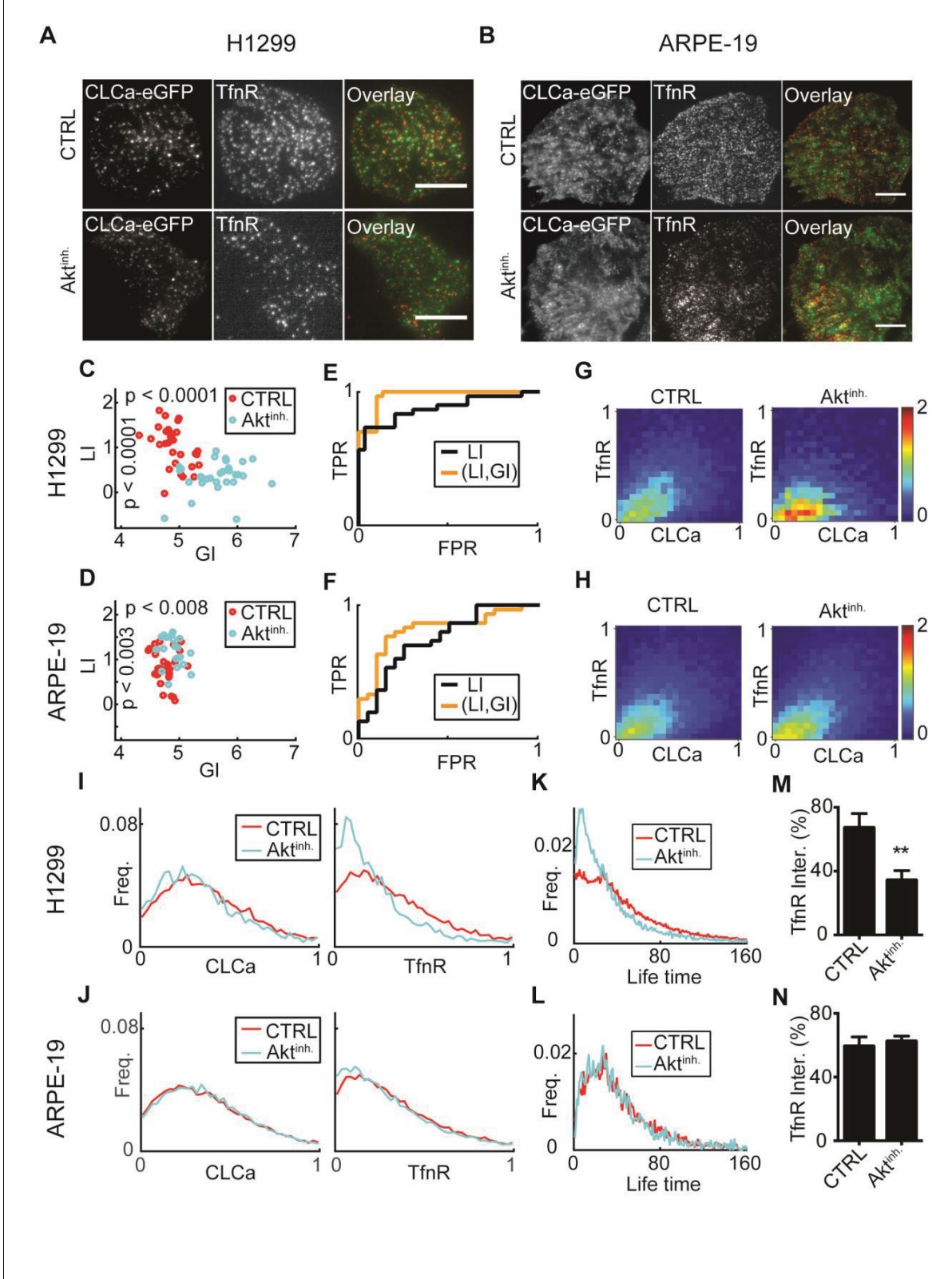

**Figure 6.** : AKT inhibition differentially alters recruitment of TfnR to CCPs during CME for different cell lines. (**A**) H1299 cells expressing CLCa and TfnR ligands. Top row, representative WT cell (TfnR ligand, GI = 4.6, LI = 1.6). Bottom row, representative AKT-inhibited cell (TfnR ligand, GI = 6.0, LI = 0.3). Scale bar 10 µm. (**B**) ARPE-19 cells. Top row, representative WT cell (TfnR ligand, GI = 4.3, LI = 0.8). Bottom row, representative AKT-inhibited cell (TfnR ligand, GI = 4.6, LI = 1.6). (**C–D**) LI and GI of CLCa-TfnR co-localization for Ctrl (red) and Akt^inh. cells (cyan). Every data point represents the LI and GI

*Figure 6 continued on next page*

*Figure 6 continued*

values for a single cell. Statistical analyses performed with the Wilcoxon rank-sum test. All DeBias analyses were performed with K = 40. (**C**) H1299: N number of cells Ctrl = 30, Akt$^{inh.}$ = 30; number of CCPs per cell: Ctrl = 455.5, Akt$^{inh.}$ = 179.5. GI p-value < 0.0001, LI p-value < 0.0001. (**D**) ARPE-19: N number of cells Ctrl = 30, Akt$^{inh.}$ = 20; number of CCPs per cell: Ctrl = 958.8, Akt$^{inh.}$ = 1138.2. GI p-value < 0.002, LI p-value < 0.008. (**E–F**) Receiver Operating Characteristic (ROC) curves showing the true positive rates as a function of false-positive rates for single cell classification, higher curves correspond to enhanced discrimination (Materials and methods). Black – LI, orange – (GI,LI). Statistics via permutation test (Materials and methods). (**E**) H1299 AUC: (GI,LI) = 0.96 versus LI = 0.88, p-value ≤ 0.003. (**F**) ARPE-19 AUC: (GI,LI) = 0.83 versus LI = 0.72, p-value ≤ 0.048. (**G–H**) Joint distributions of CLCa (x-axis) and TfnR (y-axis) for H1299 (**G**) and ARPE-19 (**H**) cells. (**I–J**) Marginal distributions of CLCa (left) and TfnR (right) for H1299 (**I**) and ARPE-19 (**J**) cells. (**K–L**) Combined CCP lifetime distribution for 50 Ctrl (red) and Akt$^{inh.}$ (cyan) cells. Statistics with Wilcoxon rank-sum test (Materials and methods). (**K**) H1299: p-value < 0.006 (mean EMD: Ctrl = 29.3 versus Akt$^{inh.}$ = 43.6); number of cells: 50 (Ctrl), 11 (Akt$^{inh.}$). (**L**) H1299: p-value n.s. (mean EMD: Ctrl = 36.1 versus Akt$^{inh.}$ = 38.0); number of cells: 12 (Ctrl), 12 (Akt$^{inh.}$). (**M–N**) Percentage of TfnR uptake: Ctrl versus Akt$^{inh.}$ (whiskers - standard deviation). Statistics via two-tailed Student's t-test. (**M**) H1299: p-value < 0.005; N = 3 independent experiments. (**N**) ARPE-19: p-value n.s.; N = 3 independent experiments.

The following figure supplement is available for figure 6:

**Figure supplement 1.** : GI encodes information that is distinct from local interactions; Experimental validations of DeBias for co-localization.

CCPs upon Akt inhibition results in an increase in short-lived, most likely abortive events, and hence a decrease in CME efficiency.

Altogether, DeBias could distinguish alterations in the regulation of CME between two cell types. The decoupling to GI and LI indicated that upon Akt inhibition, both untransformed and cancer cells showed a global bias towards CCPs with lowered TfnR intensities. This conclusion could not have been reached by considering only the LI, which increased for normal and decreased for transformed cell lines.

## Discussion

We introduce DeBias as a new method to assess global bias and local interactions between coupled cellular variables. Although the method is generic, we show here specific examples of DeBias analysis in co-orientation and co-localization studies. The source code is available, https://github.com/DanuserLab/DeBias, as well as via a web-based platform, https://debias.biohpc.swmed.edu. The website also provides detailed instructions for the operation of the user interface.

DeBias defines a generalizable framework for eliminating confounding factors in the analysis of interacting variables. Our examples demonstrate that the distinction of global and local contributions to the level of variable coupling eliminates much of the global confounder bias in the analysis of more direct interactions and can unearth in the form of global bias mechanisms that are missed by a single parameter analysis (*Figures 1–2*). In the example of vimentin-microtubule alignment (*Figure 3*), the significant decrease in GI as opposed to the LI upon partial vimentin knock-down indicated that the reduction in alignment between the two cytoskeleton systems is associated with a reduction of cell polarity as the global cue. In the example of stress-velocity alignment (*Figure 4*), depletion of some tight junction proteins increased LI, suggestive of enhanced local stress-motion transmission; knock-down of others decreased GI indicating an overall impaired alignment of velocity in the direction of wound closure. In the example of FRET experiments (*Figure 5*), PKC inhibition lead to increased LI, validating the FRET response, while a reduced GI was indicative of weaker interactions of the inactivated kinase with its substrates. In the example of Tfn receptor (TfnR) co-localization with CCPs during CME (*Figure 6*), the increased GI in response to Akt inhibition related to a higher fraction of CCPs containing less TfnR. Moreover, Akt inhibition induced opposite shifts in LI for normal and cancer cells, reflecting differential alterations in co-localization between cell types. Thus, DeBias provided insight into the regulation of cargo-pit association by kinase activity that depended on a proper deconvolution of local and global effects on the interaction of the clathrin and receptor signal. We then validated our conclusions by further analyses of the marginal distributions, live-imaging and uptake assays (*Figure 6*). Overall, the four applications shown in this work first emphasize the general need for a confounder analysis when dealing with coupled biological variables and second indicate that the global bias may be linked to mechanistically meaningful

properties of the studied system. These properties were either ignored or eliminated by previous methods, and now can be assessed directly by DeBias.

Other approaches have been used to address global confounders for assessment of local interactions between biological variables. For the specific example of object-based co-localization, Helmuth et al. simulated the spatial distribution of objects in the absence of local interactions to calibrate co-localization measurement (*Helmuth et al., 2010*). Other methods mostly used second-order spatial statistics on distances between neighbor points to exclude confounders for better co-localization sensitivity (reviewed in *Lagache et al., 2015*). Importantly, we show applications of DeBias on co-localization that do not require initial object detection (*Figure 5*). While the phenomenon of confounder bias is independent of object- versus pixel-based co-localization, we distinguish the peculiarities of the two scenarios in Appendix 4.

An important and more general approach to revealing local interactions masked by global biases was recently proposed by (*Krishnaswamy et al., 2014*), using applications to single cell mass cytometry data as examples. The authors developed a measure referred to as conditional-Density Resampled Estimate of Mutual Information (DREMI) to quantify the influence of a protein X on protein Y based on the conditional probability P(Y|X). DREMI takes advantage of the abundant mass cytometry data to equally weigh data at different intervals along the range of X values using > 10,000 cells per experimental condition. This approach is less reliable when limited data is available, because of the low confidence in the conditional probability of observations with low data abundance. Thus, DREMI is not well suited for image data, which typically has fewer observations.

DeBias estimates GI and LI assuming a constant global bias and local interaction for all observations. Moreover, its quantification power is relative. For example, a two-fold increase in the direct interaction of two variables would not necessarily result in a two-fold increase in LI. Another limitation is the absence of complete orthogonality of GI and LI values (*Figure 2E–F*, *Appendix 3—figure 1*), which complicates the interpretation of GI and LI in certain scenarios. These three limitations apply to all current approaches for quantifying interactions between coupled variables. The main conceptual advance DeBias seeks to make relates to the explicit integration of confounding factors in the analysis of coupled variables, which implies an expansion of the coupling metric from a scalar to a two-dimensional score. A forth limitation in the current implementation of DeBias is the linear normalization of multiple intensity variables in co-localization applications. Future versions may include non-linear normalization methods, although such normalization is usually highly specific to a particular data set. Last, the mechanism encoded by the GI is not always obvious. Sometimes it requires additional experiments to unveil the information contained by the GI. For example, we combined fixed cell dual-color imaging with live-imaging and uptake assays to show that shifts in the GI encode a shift in the relative populations of short- and long-lived CCPs between conditions (*Figure 6*). Despite some of the discussed complexities, DeBias offers a simple means to quantify and interpret mechanisms that alter confounders in the coupling of two variables and to largely exclude such global biases from the quantification of direct interactions.

## Materials and methods

### DeBias procedure

The DeBias procedure is depicted in *Figure 2A*. The marginal distributions X and Y are estimated from the experimental data, $\forall i, x_i, y_i \in [0, 90°]$. The experimentally observed alignment distribution (denoted *observed*) is calculated from the alignment angles $\theta_i$ of matched ($x_i,y_i$) paired variables, for all i.

$$\theta_i = \begin{cases} |x_i - y_i| & |x_i - y_i| \leq 90 \\ 180 - |x_i - y_i| & |x_i - y_i| > 90 \end{cases}$$

The resampled alignment distribution (denoted *resampled*) is constructed by independent sampling from X and Y. N random observations (where N = |X| is the original sample size) from X and Y are independently sampled with replacement, arbitrarily matched and their alignment angles calculated to define the resampled alignment. This type of resampling precludes the local dependencies between the originally matched ($x_i,y_i$) paired variables.

The uniform alignment distribution (denoted *uniform*) is used as a baseline for comparison between distributions. This is the expected alignment distribution when neither global bias (reflected by uniform X, Y distributions) nor local interactions exist. The Earth Mover's Distance (EMD) (*Peleg et al., 1989*; *Rubner et al., 2000*) was used as a distance metric between alignment distributions. The EMD for two distributions, A and B, is defined as follows:

$$EMD(A, B) = \sum_{j=1,...,K} |\sum_{j=1,...,i} a_j - \sum_{j=1,...,i} b_j|,$$ where $a_j$ and $b_j$ are the frequencies of observations in bins $j$

of the histograms of distributions A and B, respectively, each containing K bins.

The global index (GI) is defined as the EMD between the uniform distribution and the resampled alignment:

GI = EMD(*uniform,resampled*)

The local index is determined by subtraction of the global index from the EMD between the uniform distribution and the experimentally observed alignment distribution:

LI = EMD(*uniform,observed*) - global index

## DeBias for protein-protein co-localization

The following adjustments to this procedure are implemented to allow DeBias to quantify protein-protein co-localization:

1. Levels of fluorescence are not comparable between different channels due to different expression levels and imaging parameters. Thus, each channel is normalized to [0,1] by the fifth and 95th percentiles of the corresponding fluorescence intensities to achieve a stable and robust distribution.
2. The alignment angle $\theta_i$ of the matched observation $(x_i, y_i)$ is calculated as the difference in normalized fluorescence intensities $x_i$ - $y_i$ and the alignment distribution is thus defined on the interval $[-1,1]$.

The number of histogram bins for the alignment distributions (observed, resampled and uniform) was K = 15 for orientational data, 19 for PKC and 40 for CME co-localization data.

## Automated selection of number of histogram bins, K

The Freedman-Diaconis rule (*Freedman and Diaconis, 1981*) was used to automate the selection of histogram bin width: $bin\ size = \frac{Q_3(x) - Q_1(x)}{\sqrt[3]{n}}$, where $Q_i$ is the $i^{th}$ quartile of the empirical distribution x and n is the number of data points contained. A function to calculate K is included in our publicly available source code and this functionality was also integrated to the web-server implementation. Importantly, GI and LI across experiments can be compared only when evaluated with the same K value and this is enforced by the web-server. It is the responsibility of the source-code user to validate using the same K values when comparing different experimental conditions.

## Simulating synthetic data
### Simulating co-alignment data

Let us define $X$ and $Y$ as the angular probability distribution functions, with angle instances denoted $x_i$ and $y_i$, respectively. When simulating local relations, for each pair of angles, one of the angles will be shifted towards the other by $\zeta$ degrees (*Figure 1B*), unless $|x_i - y_i| < \zeta$, in which case it will be shifted by $|x_i - y_i|$ degrees. The angle to be shifted (either $x_i$ or $y_i$) is chosen by a Bernoulli random variable, $p$, with probability 0.5. The observed angles for pixel $i$ will therefore be

$$x_i' = \begin{cases} x_i & p = 1 \\ \max(x_i - \zeta, y_i) & y_i \leq x_i \wedge p = 0 \\ \min(x_i + \zeta, y_i) & y_i > x_i \wedge p = 0 \end{cases}$$

and

$$y_i' = \begin{cases} y_i & p = 0 \\ \max(y_i - \zeta, x_i) & x_i \leq y_i \wedge p = 1 \\ \min(y_i + \zeta, x_i) & x_i > y_i \wedge p = 1 \end{cases}$$

The alignment of angles at pixel $i$ will be:

$$\theta_i = \begin{cases} |x'_i - y'_i| & |x'_i - y'_i| \leq 90 \\ 180 - |x'_i - y'_i| & |x'_i - y'_i| > 90 \end{cases}$$

For example, for our simulations we choose $X, Y$ to be truncated normal distributions on $(-90, 90)$ with $\mu = 0$ and varying values of $\sigma$.

$\zeta$ is modeled in two ways: either as a constant value, e.g. $\zeta = 5°$ (*Appendix 2—figures 1–3*), or as a varying value dependent on $|x_i - y_i|$ (*Figures 1–2*). For the latter, $\zeta$ is defined as a fraction $0 < \alpha < 1$ from $|x_i - y_i|$ for each pair of observations; namely, $\zeta_i = \alpha |x_i - y_i|$ (see *Figure 1B*). Note, that the observed marginal distributions X', Y' may be slightly different from X, Y.

## Simulating co-localization data

Let us define $X$ as a probability distribution function, with instances denoted $x_i$. Local interactions were simulated as $y_i = x_i \zeta_i$, where $\zeta_i$ is an instance of a probability distribution $Z$. For our simulations we chose $X$ to be a truncated normal distribution on $[0, 1]$ with $\mu_x = 0.5$ and $Z$ to be a normal distribution with $\mu_\zeta = 1$. This model of interaction assumes on average a one-to-one interaction between $X$ and $Y$, deviation of $\zeta_i$ from one implies reduced interaction. $y_i$ samples were also truncated to $[0,1]$. This ensures that $\forall i, 0 \leq y_i, x_i \leq 1$ and accordingly, $\forall i, -1 \leq x_i - y_i, \leq 1$, making unnecessary the normalization step in DeBias co-localization calculation.

When simulating scenarios where only sub-groups of the observations undergo interactions, we sampled the none-interacting observations $y_i$ from $Y$ the truncated normal distribution on $(0, 1)$, with $\mu_y = 0.5$ and $\sigma_y = \sigma_x$ (same as $X$).

# Vimentin and microtubule filaments experiments and analysis

## Cell model

hTERT-RPE-1 cells (ATCC, RRID: CVCL_4388) were TALEN-genome edited to endogenously label vimentin with mEmerald and α-tubulin with mTagRFPt, and validated for protein expression levels (*Gan et al., 2016*). Cells were stably transfected with shRNA against vimentin to knock down vimentin and the knockdown efficiency was validated as ~75% (*Gan et al., 2016*). The cell line has been tested negative for mycoplasma contamination.

## Fixed cell imaging

hTERT-RPE-1 mEmerald-vimentin/mTagRFPt-α-tubulin cells expressing shRNA-VIM or scrambled control shRNA Scr were plated into MatTek (Ashland, MA) 35 mm glass-bottom dishes (P35G-0–20 C) coated with 5 µg/mL fibronectin. Cells were incubated overnight to allow them to adhere and form monolayers. Monolayers were scratched with a pipette tip to form a wound. Cells were incubated for 90 min, washed briefly and fixed with methanol at −20°C for 15 min. Cells were imaged at the wound edge (denoted 'front' cells), and at 2–3 cell rows from the wound edge (denoted 'back' cells, only for control condition). Images were acquired using a Nikon Eclipse Ti microscope, equipped with a Nikon Plan Apo Lambda 100x/1.45 N.A. objective. Images were recorded with a Hamamatsu ORCA Flash 4.0 with 6.45 µm pixel size (physical pixel size: 0.0645 × 0.0645 µm). All microscope components were controlled by Micro-manager software.

## Live cell imaging

hTERT-RPE-1 mEmerald-vimentin/mTagRFPt-α-tubulin cells expressing scrambled control shRNA Scr were plated into MatTek (Ashland, MA) 35 mm glass-bottom dishes (P35G-0–20 C) coated with 5 µg/mL fibronectin. Cells were incubated overnight to allow them to adhere and form monolayers. Monolayers were scratched with a pipette tip to form a wound. Imaging started 30 min after scratching with an Andor Revolution XD spinning disk microscope mounted on a Nikon Eclipse Ti stand equipped with Perfect Focus, a Nikon Apo 60 × 1.49 N.A. oil objective and a 1.5x optovar for further magnification. Images were recorded with an Andor IXON Ultra EMCCD camera with 16 µm pixel size (physical pixel size: 0.16 × 0.16 µm). Lasers with 488 nm and 561 nm light emission were used for exciting mEmerald and mTagRFPt, respectively. The output powers of the 488 nm and 561 nm lasers were set to 10% and 20% of the maximal output (37 mW and 23 mW, respectively). The exposure time was 300 ms per frame for both channels and images were collected at a frame rate of

1 frame per minute. During acquisition, cells were kept in an onboard environmental control chamber. All microscope components were controlled by Metamorph software.

## Filament extraction and spatial matching

We applied the filament reconstruction algorithm reported in (*Gan et al., 2016*). Briefly, multi-scale steerable filtering is used to enhance curvilinear image structures, centerlines of candidate filament fragments are detected, clustered to high and low confidence sets and iterative graph matching is applied to connect fragments into complete filaments. Each filament is represented by an ordered chain of pixels and the local filament orientation derived from the steerable filter response. Spatial matching was performed as follows: each pixel belonging to a filament detected in the MT channel is recorded to the closest pixel that belongs to a filament in the VIM channel. If the distance between the two pixels is less than 20 pixels, then the pair of VIM and MT orientations at this pixel is recorded for analysis. The same process is repeated to record matched pixels from VIM to MT filaments.

## Collective cell migration experiments and analysis

Coupled measurements of velocity direction and stress orientation were taken from the data originally published by Tamal Das et al. (*Das et al., 2015*). Particle image velocimetry (PIV) was applied to calculate velocity vectors, and monolayer stress microscopy (*Tambe et al., 2011*) was used to extract stress orientations. Velocity and stress measurements were recorded 3 hr after collective migration was induced by lifting off the culture-insert in which the cells have grown to confluence. Validated siRNAs were used for gene screening. Detailed experimental settings can be found in *Das et al., 2015*.

## Statistical test

We devised a permutation test to determine statistical significance of differences in LI values between experiments (conditions) (*Figure 4C*). For every pair of conditions (i,j), the following procedure was repeated for 100 iterations. 50% of the velocity-stress observations were randomly selected for each condition and the LI (and GI) were calculated from this subsampling. Without loss of generality, for $LI_i < LI_j$ (based on *Figure 4B*) the p-value was recorded as the fraction of iterations in which the subsampled LI value for condition i was greater than the LI value for condition j. A fraction of 0 thus implies p-value < 0.01.

## PKC FRET experiments

hTERT-RPE-1 cells (ATCC, RRID: CVCL_4388) expressing GFP and mCherry (for the control experiment) or C kinase activity reporter (CKAR, for the PKC activation experiment) (*Violin et al., 2003*) were plated with DMEM/F12 medium containing 10% fetal bovine serum and 1% penicillin-streptomycin in fibronectin-coated 35 mm MatTek plates (P35G-0–10 c). The cell line has been tested negative for mycoplasma contamination. Cells were incubated overnight and imaged with a custom-built DeltaVision OMX widefield microscope (GE healthcare life sciences, Chicago, IL) equipped with an Olympus PLAN 60 × 1.42 N.A. oil objective and CoolSNAP HQ2 interline CCD cameras. FRET experiments were performed with a 445 nm laser, and control experiments were performed with 488 and 561 nm lasers. 478/35, 541/22, 528/48 and 609/37 emission bandpass filters were used for the CFP, YFP, GFP and mCherry channels, respectively. The output powers of the lasers were set to 10% the maximal output (100 mW). The exposure time was 200 ms per frame for both channels. 10 µM of the PKC inhibitor HA-100 dihydrochloride (Santa Cruz Biotechnology, Dallas, TX) was added to the media after the first image was recorded and a second image was recorded 20 min later.

Single cells were manually selected for analysis. In particular, cells with higher intensities in the CFP channel were found to provide reproducible changes in their FRET intensity. Each cell was manually annotated and analyzed with the Biosensor Processing Software 2.1 to produce the ratio images (*Hodgson et al., 2010*). Briefly, the field of view was corrected for uneven illumination, background was subtracted, the image was masked with the single cell annotation, and the ratio image was calculated as $CFP/YFP_{CFP}$. Statistics was determined using the non-parametric Wilcoxon signed-rank test.

## Clathrin mediated endocytosis experiments

### Cells, cell culture and chemicals

ARPE-19 (retinal pigment epithelial cells) (ATCC, RRID: CVCL_0145) stably expressing eGFP-CLCa were grown under 5% CO2 at 37°C in DMEM high glucose medium (Life Technologies, ), supplemented with 20 mM HEPES, 10 mg/ml streptomycin, 66 ug/ml penicillin and 10% (v/v) fetal calf serum (FCS, HyClone). H1299 (non-small cell lung carcinoma) (RRID: CVCL_0060, a generous gift from Dr. J. Minna at the UT Southwestern Medical Center) stably expressing eGFP-CLCa were grown under 5% $CO_2$ at 37°C in RPMI, supplemented with 20 mM HEPES, 10 mg/ml streptomycin, 66 ug/ml penicillin and 5% (v/v) fetal calf serum (FCS, HyClone). STR profiling was performed to ensure cell identity. No mycoplasma contamination was found. The AKT inhibitor (Akt inhibitor X, 'ten') was purchased from Calbiochem.

### Transferrin receptor internalization

TfnR uptake was measured by an 'in-cell' ELISA assay using the anti-TfnR monoclonal antibody, HTR-D65 (*Schmid and Smythe, 1991*), as ligand, exactly as previously described (*Reis et al., 2015*). Internalized D65 was expressed as the percentage of the total surface-bound D65 at 4°C (i.e., without acid wash step), measured in parallel.

### Fixed cell imaging

Transferrin receptor (TfnR) surface levels were measured using the anti-TfnR mAb (HTR-D65). ARPE-19 and H1299 cells (1.5 × 105 cells per well in a 6-well plate) were grown overnight on glass cover slips and further pre-incubated with 4 ug/ml of D65 in TfnR assay buffer (PBS4+: PBS supplemented with 1 mM MgCl2, 1 mM CaCl2, 5 mM glucose and 0.2% bovine serum albumin) at 4°C for 30 min. After being washed with PBS$^{4+}$, cells were fixed in 4% PFA for 30 min at 37°C, permeabilized with 0.1% Triton X-100 for 5 min and further blocked with Q-PBS (2% BSA, 0.1% lysine, pH 7.4) for 30 min. After three washes with PBS, cells were incubated with a 1:500 dilution of goat anti-mouse Alexa-568 labelled secondary antibody (Life Technologies) for 30 min, washed an additional three times with PBS before TIRFM imaging using a 100 × 1.49 NA Apo TIRF objective (Nikon, Japan) mounted on a Ti-Eclipse inverted microscope equipped with the Perfect Focus System (Nikon). Images were acquired with an exposure time of 150 ms for both channels using a pco-edge 5.5 sCMOS camera with 6.5 um pixel size. For inhibition studies, cells were initially pre-incubated in the presence of Akt inhibitor X (10 uM) for 30 min at 37°C, followed by pre-incubation with 4 ug/ml of D65 at 4°C for 30 min, in continued presence of the inhibitor.

### Live-cell imaging and analysis

During TIRFM imaging, cells were maintained in DMEM lacking phenol red and supplemented with 2.5% fetal calf serum. Time-lapse image sequences were acquired at a frame rate of 1 frame/s with exposure time of 150 ms using a pco-edge 5.5 sCMOS camera with 6.5 um pixel size. CCP detection, tracking and construction of life-time distributions were performed with the custom CME analysis software described in (*Aguet et al., 2013*). Lifetime distribution was defined at the resolution of 1 s and limited to 160 s. Longer CCP trajectories were excluded from the analysis. To compare lifetime distributions for single field of views (FOVs) between WT and AKT-inhibited cells we measured the heterogeneity as the EMD distance to the uniform distribution. FOV's score were compared between the different experimental conditions using the non-parametric Wilcoxon rank-sum test.

### Image analysis for fixed cell experiments

Single cell masks were manually annotated in each FOV. We applied the approach described in (*Aguet et al., 2013*) to automatically detect CCPs from the CLC channel. Briefly, CLC fluorescence was modeled as a two-dimensional Gaussian approximation of the microscope PSF above a spatially varying local background. CCP candidates were first detected via filtering, followed by a model-fitting for sub-pixel localization. The fluorescent intensity of the CLC and any other acquired channel were recorded in the detection coordinates to define the matched observations for DeBias. GI and LI were calculated independently for each single cell. Linear Discriminant Analysis (LDA) classification (*Fisher, 1936*) was applied to assess single cell classification accuracy. Every cell constituted an observation, a label was assigned based on the experimental condition and the representation was

either the LI or the pair (GI,LI). The LDA classifier was trained on a labeled dataset consisting of WT and AKT-inhibition for H1299 or ARPE19 cells. The area under the Receiver Operating Characteristic (ROC) curve was recorded to assess and compare the discriminative accuracy of different measures. The true-positive rate (TPR) is the percentage of control cells classified correctly. The false-positive rate (FPR) is the percent of AKT inhibited cells classified as control. When comparing the potential accuracy of several classification algorithms, a measure that has higher true-positive rate for any fixed false-positive rate values is proved to be the better one. Thus, higher curves (larger areas under the ROC curve, or AUC) correspond to more discriminative measures. Statistical significance for comparing classification performance of LDA classifiers that were trained for scalar measures with or without the GI was calculated by bootstrapping (*Figure 6E–F*). The following process was repeated 1000 times and the frequency for which the scalar-based classifier outperformed the classifier trained on pairs of measures was reported as the p-value. Random resampling with replacement was performed to obtain a sample size identical to that of the observed dataset. The area under the curve (AUC) of the competing pre-trained LDA classifiers was assessed for this resampled dataset and recorded when the model that was trained without the GI predicted better.

## Webserver

The DeBias code was implemented in Matlab, compiled with Matlab complier SDK and transferred to a web-based platform to allow public access for all users at https://debias.biohpc.swmed.edu. The graphical user interface (GUI) was designed to be simple and easy to use. The user uploads one or more datasets to the DeBias webserver and selects the mode of operation (co-localization/orientation). GI and LI values are calculated and the results are displayed and emailed to the user. 'DeBias Analyst' enables to group experiments into two experimental conditions (usually control versus treatment), visualizes and outputs statistics on the alterations of GI and LI. The software's flow chart and a detailed user manual are available in the online user manual. Source code is publicly available, https://github.com/DanuserLab/DeBias *Danuser, 2017* (with a copy archived at https://github.com/elifesciences-publications/DeBias).

## Acknowledgements

We thank the BioHPC team at UTSW and especially Liqiang Wang for the help in implementing the DeBias web-server and making it freely available for public use. We are grateful to Tamal Das and Joachim Spatz for providing us with the motion and stress data from their tight-junction screen. We thank Andrew Jamieson for helping to package the source code and creating the repository. We thank Dana Reed for assistance with cell characterization and mycoplasma testing. We thank Sangyoon Han, Claudia Schaefer, Meghan Driscoll, Erik Welf, Phillippe Roudot and Marcel Mettlen for critically reading the manuscript and Marcel Mettlen for fruitful discussions and advice. This work was supported by the Cancer Prevention and Research Institute of Texas (CPRIT R1225 to GD), by NIH P01 GM103723, P01 GM096971 (to GD) and by GM713165 (to SLS and GD). UO was supported by an EMBO Long-Term fellowship. The funders had no role in study design, data collection and analysis, decision to publish or preparation of the manuscript.

## Additional information

### Funding

| Funder | Grant reference number | Author |
|---|---|---|
| EMBO | postdoctoral fellowship | Uri Obolski |
| National Institute of General Medical Sciences | R01 GM073165 | Gaudenz Danuser Sandra L Schmid |
| Cancer Prevention and Research Institute of Texas | R1225 | Gaudenz Danuser |
| National Institute of General Medical Sciences | P01 GM103723 | Gaudenz Danuser |
| National Institute of General | P01 GM096971 | Gaudenz Danuser |

Medical Sciences

The funders had no role in study design, data collection and interpretation, or the decision to submit the work for publication.

## Author contributions

AZ, Conceptualization, Software, Formal analysis, Investigation, Methodology, Writing—original draft, Writing—review and editing; UO, Formal analysis, Investigation, Methodology, Writing—review and editing; ZG, CRR, Investigation, Writing—review and editing; ZK, Helped in troubleshooting and optimizing the analysis, Assisted with preparing the manuscript; YD, Resources, Software; SLS, Supervision, Writing—review and editing; GD, Conceptualization, Supervision, Writing—review and editing

## Author ORCIDs

Assaf Zaritsky, http://orcid.org/0000-0002-1477-5478
Zhuo Gan, http://orcid.org/0000-0002-7738-9072
Sandra L Schmid, http://orcid.org/0000-0002-1690-7024
Gaudenz Danuser, http://orcid.org/0000-0001-8583-2014

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

## Appendix 1

## Theoretical results for co-orientation data

### Terms and definitions

Let X, Y be the distribution functions of two random variables representing angles on $[-90°, 90°]$. Spatially matched random variables from these distributions are denoted $x_i$ and $y_i$, $i = 1, 2 \ldots N$, where $N$ is the number of observations. $x_i^*$ and $y_i^*$ are random variables sampled from X and Y independently (without considering the spatial matching). The observed and resampled alignment distributions are denoted $A$ and $A^*$, respectively. The alignment distributions represent angles on $[0°, 90°]$, and are functions of $X$, $Y$, the interaction between $X$ and $Y$, and $N$. Random variables from $A$ are denoted $\theta_i$, $i = 1, 2 \ldots N$. Random variables from $A^*$ are denoted $\theta_i^*$. Let $K$ be the number of bins in the alignment histogram. Histogram bins are denoted $bin_i$, $i = 0, \ldots, K-1$ where $bin_0$ contains the lowest values (including 0°) and $bin_{k-1}$ the highest values (including 90°). $U$ denotes the uniform distribution on $[0°, 90°]$ with the same K bins as $A$, $A^*$.

### Theorem 1: Local index of independent variables

If $X$, $Y$ are independent, then $E(LI(X, Y)) = 0$

Proof:

$$
\begin{aligned}
E(LI) &= E(EMD(A, U) - EMD(A^*, U)) \\
&= E(EMD(A, U)) - \underbrace{E(EMD(A^*, U))}_{*} = E(EMD(A, U)) - E(EMD(A, U)) = 0
\end{aligned}
$$

\* Resampling does not change the expectation of the difference of two independent variables.

### Theorem 2: Global index of uniform distributions

Let $X, Y$ be uniform distributions, then $\lim_{N \to \infty} GI = 0$.

Proof:

Since $X$ and $Y$ are uniform distributions, the density of the resampled distributions are $f_{x_i^*}(\omega) = f_{y_i^*}(\omega) = \frac{1}{180}, -90 < \omega < 90$. We can compute the distribution of the difference by

$$
f_{x_i^* - y_i^*}(\omega) = \int_{-\infty}^{\infty} f_{x_i^*}(x) f_{y_i^*}(x - \omega) dx = \int_{-\infty}^{\infty} \frac{1}{180^2} I_{x \in (-90, 90)} I_{x - \omega \in (-90, 90)} dx =
$$

$$
\begin{cases}
\frac{1}{180^2} \int_{-90}^{90 + \omega} 1 dx & 0 < \omega < 180 \\
\frac{1}{180^2} \int_{-90 + \omega}^{90} 1 dx & 0 < \omega < 180
\end{cases}
=
\begin{cases}
\frac{1}{180^2}(180 + \theta) & -180 < \omega < 0 \\
\frac{1}{180^2}(180 - \theta) & 0 < \omega < 180
\end{cases}
$$

We conclude that the distribution of the difference is

$$
f_{x_i^* - y_i^*}(\omega) =
\begin{cases}
\frac{1}{180}\left(1 + \frac{\omega}{180}\right) & -180 < \omega < 0 \\
\frac{1}{180}\left(1 - \frac{\omega}{180}\right) & 0 < \omega < 180
\end{cases}
$$

Therefore, when taking the absolute value of the angle difference we get:

$$f_{|x_i^* - y_i^*|}(\omega) = \begin{cases} \frac{1}{90}(1 - \frac{\omega}{180}) & \omega \geq 0 \\ 0 & \omega < 0 \end{cases}$$

Finally, since the alignment is limited to $[0°, 90°]$ we apply the function

$$g(\omega) = \begin{cases} 180 - \omega & 90 < \omega \leq 180 \\ \omega & 0 < \omega \leq 90 \end{cases} \quad \text{so that}$$

$$f_{g(|x_i^* - y_i^*|)}(\omega) = \begin{cases} \frac{1}{90}(1 - \frac{w}{180}) + \frac{1}{90}(1 - \frac{180 - \omega}{180}) & 0 \leq \omega \leq 90 \\ 0 & else \end{cases}$$

$$= \begin{cases} \frac{1}{90} & 0 \leq \omega \leq 90 \\ 0 & else \end{cases}$$

Therefore $f_{g(|x_i^* - y_j^*|)} = A^*$ is a uniform distribution.

For $N$ sufficiently large, the histogram has approximately $\frac{1}{K}$ of the observations in each of the $K$ equally spaced intervals between 0 and 90 and thus $\lim_{N \to \infty} GI = \lim_{N \to \infty} EMD(A^*, U) = 0$.

## Theorem 3: Perfect alignment

1. If $\forall i, j, \ x_i = y_j$ then $GI = \frac{(K-1)}{2}, \ \lim_{N \to \infty} LI = 0$

2. If $X$ and $Y$ are uniform distributions, and $\forall i, \ x_i = y_i$ then $\lim_{N \to \infty} GI = 0, \lim_{N \to \infty} LI = \frac{(K-1)}{2}$

Proof:

(I)

$A = A^*$ because $\forall i, j \ a_i = a_j^* = 0$. For large $N$ the random variable drawn from the alignment distribution $A$ will be approximately:

$$\theta_i = \begin{cases} 1 & \theta_i \in bin_0 \\ 0 & else \end{cases}, \forall i.$$

The EMD of the alignment from the uniform distribution is therefore simply 'moving' $\frac{1}{K}$ observations from $bin_0$ to every other bin, which sums up to

$$\lim_{N \to \infty} EMD(A, U) = \frac{1}{K} * 1 + \frac{1}{K} * 2 + \ldots + \frac{1}{K} * (K-1) = \frac{1}{K} * (K-1) * \frac{1 + K - 1}{2} = \frac{K-1}{2}$$

Therefore, for large $N$, $LI = EMD(A, U) - EMD(A^*, U) = 0$, $GI = EMD(A^*, U) = \frac{K-1}{2}$.

(II)

Since $\forall i, \ x_i = y_i$ we get that, similarly to part (I), $EMD(A, U) = \frac{K-1}{2}$.

On the other hand, since $X$ and $Y$ are uniform distributions, we get from theorem 2 that $\lim_{N \to \infty} EMD(A^*, U) = 0$.

Therefore, for infinite observations, $LI = \frac{K-1}{2}$, $GI = 0$.

## Theorem 4: $LI$ is a lower bound for the local contribution to the observed alignment

Assuming that the observed alignment distribution $A$ is cumulatively explained by a global bias and a local interaction, we construct a new alignment distribution $A_{-\zeta}$ encoding the true cumulative local contribution to the observed alignment and demonstrate that $LI \leq$

$EMD(U, A_{-\zeta}) - GI$ to conclude that $LI$ is a lower bound for the local contribution to the observed alignment.

Proof:

We first define $A^-$, the alignment distribution corresponding to $A$ that does not include any local interaction. Thus, $A^-$, can be interpreted as an alignment distribution constructed from $X^-$ and $Y^-$, denoting $X$ and $Y$ after elimination of the (unknown) alignment correction due to local interactions between the observations $(x_i, y_i)$. The construction of $A_{-\zeta}$ is based on the corresponding matching pairs $(x_i^- \in X^-, y_i^- \in Y^-)$ with alignment correction by the local interaction $\zeta_i$ (see **Figure 1B** for as a schematic depiction). Such local interaction exists in our model (although it might not be explicitly known) and can be represented as a vector $\zeta \in \mathbb{R}^N$, $\zeta_i \geq 0 \ \forall i$. Note, that this construction supports different $\zeta_i$ values for every observation $i$ and thus can provide a more detailed platform than the single measure LI that DeBias outputs (which assumes $\zeta_i = \zeta_j \ \forall i, j$). Also note, that when $\zeta_i > \theta_i^-$ ($\theta_i^-$ is the alignment angle between $(x_i^-, y_i^-)$), then the observed alignment $\theta_i^- - \zeta_i < 0$.

Accordingly, $A_{-\zeta}$ is defined as the alignment distribution of $\theta_i^- - \zeta_i$. As described above, $A_{-\zeta}$ can contain negative values for $\zeta_i > \theta_i^-$. A, the experimentally observed alignment, thus can be generated from $A_{-\zeta}$ as well, by truncating the 'saturated' observations (where $\zeta_i > \theta_i^-$) to the value 0. More formally, the elements in $A$ are defined by

$$
\begin{array}{ll}
\theta_i^- - \zeta_i & \theta_i^- > \zeta_i \\
0 & \theta_i^- \leq \zeta_i
\end{array}
$$

We can get an upper bound for $EMD(A, A^-)$ in the form of:

$$EMD(A, A^-) \leq EMD(A_{-\zeta}, \ A^-) \leq \sum_{i=1}^{N} \frac{1}{N} \left\lceil \frac{\zeta_i}{|bin|} \right\rceil.$$

Where $|bin|$ defines the size of the angular interval of a bin in the alignment histogram.

This equation is intuitively interpreted as every observation $i$ is locally aligned by $\zeta_i$, and therefore is translocated $\left\lceil \frac{\zeta_i}{|bin|} \right\rceil$ bins, at most.

Note that a decreased bin size reduces this bound as close as needed to the value of $EMD(A_{-\zeta}, \ A^-)$.

Finally,

$$LI \underbrace{\leq}_{*} EMD(A, A^*) \approx EMD(A, \ A^-) \leq EMD(A_{-\zeta}, \ A^-) \leq \sum_{i=1}^{N} \frac{1}{N} \left\lceil \frac{\zeta_i}{|bin|} \right\rceil$$

Thus, the LI is a lower bound on the contribution of the direct interaction between $X$ and $Y$ on the alignment distribution.

Additionally, we get that

$$GI = EMD(A^*, U) = EMD(A, U) - LI \underbrace{\geq}_{*} EMD(A, U) - EMD(A^*, A)$$

$$\geq EMD(A, U) - \sum_{i=1}^{N} \frac{1}{N} \left\lceil \frac{\zeta_i}{|bin|} \right\rceil$$

Implying that the GI is an upper bound of the contribution of the global bias.

\* by corollary 2

**Corollary 2:** For any alignment distribution A, $LI \leq EMD(A, A^*)$

Proof:

Let $A_i$, $A_i^*$, $U_i$ denote the relative frequency of observations in $bin_i, 0 \leq i \leq k-1$ for $A$, $A^*$, $U$, respectively.

$$
\begin{aligned}
EMD(A, A^*) &= \sum_{0 \leq i \leq K-1} |A_i - A_i^*| \\
&= \sum_{0 \leq i \leq K-1} |A_i - U_i + U_i - A_i^*| \underset{*}{\geq} \sum_{0 \leq i \leq K-1} \left( |A_i - U_i| - |A_i^* - U_i| \right) \\
&= \sum_{0 \leq i \leq K-1} |A_i - U_i| - \sum_{0 \leq i \leq K-1} |A_i^* - U_i| = EMD(A, U) - EMD(A^*, U) = LI
\end{aligned}
$$

* triangle inequality

## Theorem 5: GI limits for highly variant truncated normal distributions

$\lim_{\substack{\sigma \to \infty \\ N \to \infty}} GI = 0$ for the following scenarios:

1. The resampled alignment is a truncated normal distribution with variance parameter $\sigma^2$.
2. $X$ and $Y$ are truncated normal distributions, each with variance parameter $\sigma^2$.

Proof:

(I)

Let $A_\sigma^*$ be the truncated normal resampled alignment distribution, defined by the parameters $\mu = 0$, $\sigma$, with the support interval $(a, b)$, such that $a \leq \mu \leq b$. Let $\phi^\sigma(x)$ be the probability density function (PDF) of the truncated normal distribution and $u(x)$, $U$ respectively, the PDF and CDF (cumulative distribution function) of the uniform distribution function on $(a, b)$. The PDF and CDF of the normal distribution function is denoted in the standard notation of $\phi$ and $\Phi$ respectively.

First we prove that $\lim_{\sigma \to \infty} \phi^\sigma(x) = u(x)$ and use this to conclude that $\lim_{\substack{\sigma \to \infty \\ N \to \infty}} EMD(A_\sigma^*, U) = \lim_{\substack{\sigma \to \infty \\ N \to \infty}} GI = 0$.

$$
\forall x_1, x_2 \in (a, b), \ \lim_{\sigma \to \infty} \frac{\phi^\sigma(x_1)}{\phi^\sigma(x_2)} = \lim_{\sigma \to \infty} \frac{\frac{\phi\left(\frac{x_1}{\sigma}\right)}{\sigma\left(\Phi\left(\frac{b}{\sigma}\right) - \Phi\left(\frac{a}{\sigma}\right)\right)}}{\frac{\phi\left(\frac{x_2}{\sigma}\right)}{\sigma\left(\Phi\left(\frac{b}{\sigma}\right) - \Phi\left(\frac{a}{\sigma}\right)\right)}} = \lim_{\sigma \to \infty} \frac{\phi\left(\frac{x_1}{\sigma}\right)}{\phi\left(\frac{x_2}{\sigma}\right)} = \lim_{\sigma \to \infty} \frac{e^{-\frac{x_1^2}{2\sigma^2}}}{e^{-\frac{x_2^2}{2\sigma^2}}}
$$

$$
= \lim_{\sigma \to \infty} e^{\frac{x_2^2 - x_1^2}{2\sigma^2}} = 1
$$

Therefore, $\lim_{\sigma \to \infty} \phi_t^\sigma(x) = Constant$. Since the support of $\phi_t^\sigma$ is $(a, b)$, the only constant satisfying that $\lim_{\sigma \to \infty} \phi_t^\sigma(x)$ is the probability distribution, $\frac{1}{b-a} = u(x)$. Therefore, $\lim_{\substack{\sigma \to \infty \\ N \to \infty}} EMD(A_\sigma^*, U) = \lim_{\substack{\sigma \to \infty \\ N \to \infty}} GI = 0$.

(II)

Let $X$, $Y$ be truncated normal distributions. In part (I) we prove that $\lim_{\sigma \to \infty} X = \lim_{\sigma \to \infty} Y = u(x)$. Theorem 2 implies that when $X$ and $Y$ are uniform distributions $\lim_{\substack{\sigma \to \infty \\ N \to \infty}} GI = 0$.

## Appendix 2

### Simulations with constant $\zeta$

To assess the performance of DeBias we tested its ability to retrieve a pre-determined local interaction parameter $\zeta$ (see **Figure 1B**) from simulated synthetic data. $X$ and $Y$ were modeled as truncated normal distributions on $(-90, 90)$, with $\mu=0$ and changing $\sigma_x, \sigma_y$. Pairs of $(X_i, Y_i)$ were sampled from $X, Y$ and shifted towards each other by $\zeta$ degrees (similar to **Figure 1B**, but with a constant cumulative $\zeta$) to construct the observed alignment angles. To avoid confusion we denote $X, Y, \sigma_x, \sigma_y$ as the observed values post-simulation. For a given constant $\zeta$, we exhaustively explored the $\sigma_x, \sigma_y$ space. For each $\sigma_x, \sigma_y$, we performed 20 independent simulations with N = 1600 observations $(X_i, Y_i)$. For each simulation we constructed the resampled distribution 10 times based on 400 observations drawn from the marginal $X, Y$ distributions, and used the mean GI, LI. The final recorded GI, LI were averaged over the independent simulations.

The expected mean alignment when neither global bias nor local interactions exist is 45°. We begin by examining the deviation of the mean observed alignment from this value (45° - $\theta_{mean}$). Better alignment is reflected by higher 45° - $\theta_{mean}$ values implying a larger deviation from the unbiased and no-interactions scenario. Low standard deviations $\sigma$, correspond to better alignment, improving with growing $\zeta$, as expected (**Appendix 2—figure 1A**). The GI follows a similar pattern and remains relatively stable for small changes in $\zeta$ (**Appendix 2—figure 1B**). The similar patterns between **Appendix 2—figure 1A and B** indicate that the global bias has a prominent role in determining the observed alignment.

The LI grows with $\zeta$ (**Appendix 2—figure 1C**) and its relative contribution to the observed alignment grow with increasing $\zeta$ (**Appendix 2—figure 1D**, quantified by LI/(LI+GI)), as expected. This relative contribution can be harnessed to restore an estimated $\zeta$ as the corresponding fraction from (45° - $\theta_{mean}$) (**Appendix 2—figure 1E**). The estimated $\zeta$ is a lower bound for the actual value (Appendix 1, Theorem 4). Estimation is more accurate for larger $\zeta$ and for large $\sigma$ (**Appendix 2—figure 1F**). These results again highlight the importance of exploiting the GI for better interpretation of the LI (first introduced in **Figure 2D–E**).

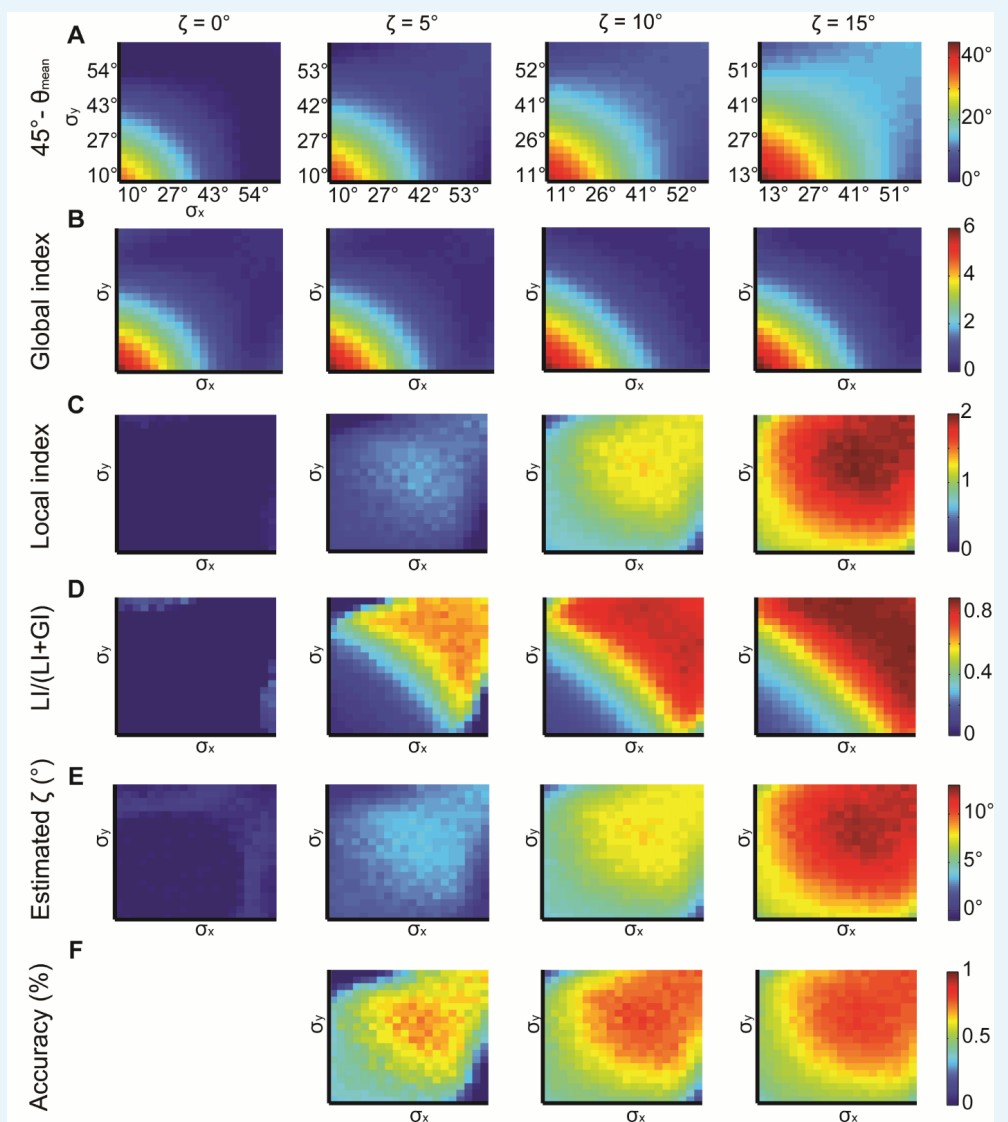

**Appendix 2—figure 1.** Simulations for X, Y normal distributions with different $\sigma_x$, $\sigma_y$ and constant $\zeta = 0°$, 5°, 10°, 15°. (**A**) 45° – $\theta_{mean}$ reflecting the cumulative effect of the global bias and the local interaction between X, Y ($\theta_{mean}$ is the mean observed alignment). Lower variance and higher $\zeta$ correspond to better alignment. (**B**) Global index. $\zeta$ has a small effect on GI. (**C**) Local index. $\zeta$ has a major effect on LI. (**D**) Relative contribution of LI to the observed alignment increases as function of $\zeta$. (**E**) Retrieved estimated $\zeta$ calculated as the relative contribution of LI to the observed alignment (panel D) times the cumulative effect of the global bias and the local interaction (panel A). (**F**) Accuracy of estimated $\zeta$ grows with $\zeta$ and with lower $\sigma_x$, $\sigma_y$. Note, that this estimation is a lower bound for the true $\zeta$ (Appendix 1, Theorem 4). Accuracy cannot be measured for $\zeta = 0°$ hence the empty panel.

We also investigated the effect of the choice of the number of bins K, used for sampling and computation of the EMD between distributions. Increased K induces linear growth in LI and GI values, as expected (Appendix 1 Theorem 5, **Appendix 2—figure 2A–B**) and stabilized its accuracy in predicting $\zeta$ for K $\geq$ 11 (**Appendix 2—figure 2C–E**). Large K will require more observations to estimate the true distribution. Using a constant K for a specific application assures fair comparison between different cases. Varying N, the number of observations, did

not have a major effect on these measurements (*Appendix 2—figure 3A–D*), but increasing N reduced the noise which increased the accuracy in predicting $\zeta$ (*Appendix 2—figure 3E*).

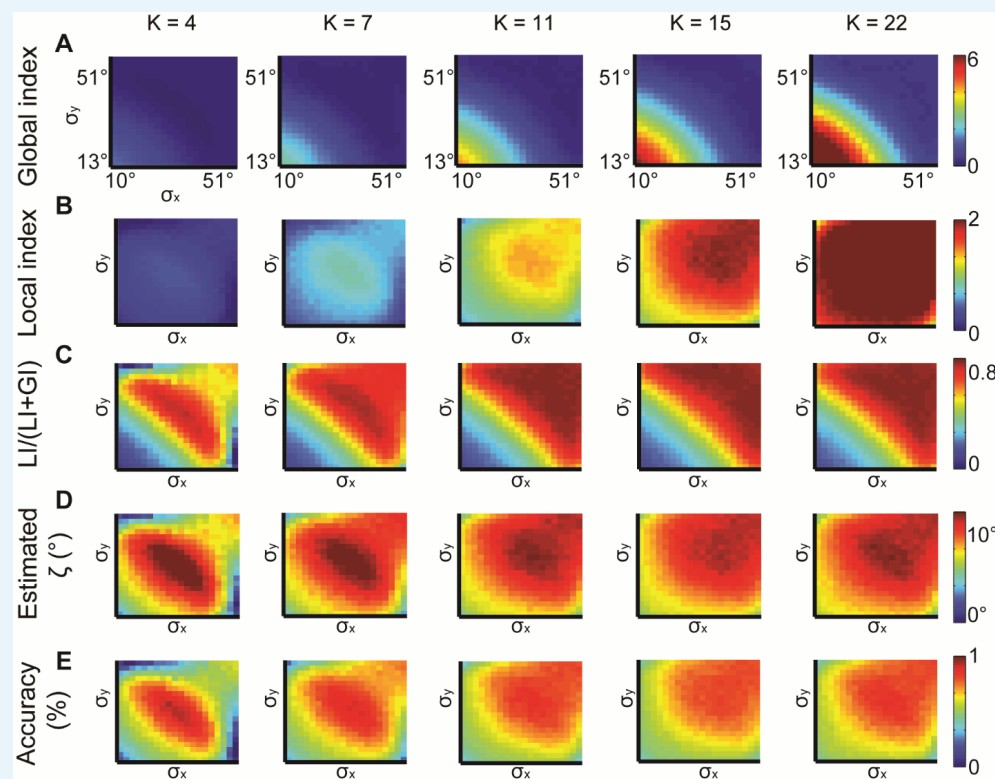

**Appendix 2—figure 2.** Simulations for different values of K, the number of bins in the alignment distribution. $X, Y$ normal distributions with different $\sigma_x, \sigma_y$ and constant $\zeta = 15°$. K = 4, 7, 11, 15, 22 were examined. (**A–B**) Global (**A**) and local (**B**) indices grow with K. (**C–E**) Relative contribution of LI to the observed alignment (**C**), Retrieved estimated $\zeta$ (**D**) and accuracy of estimated $\zeta$ (**E**) stabilizes for K $\geq$ 11.

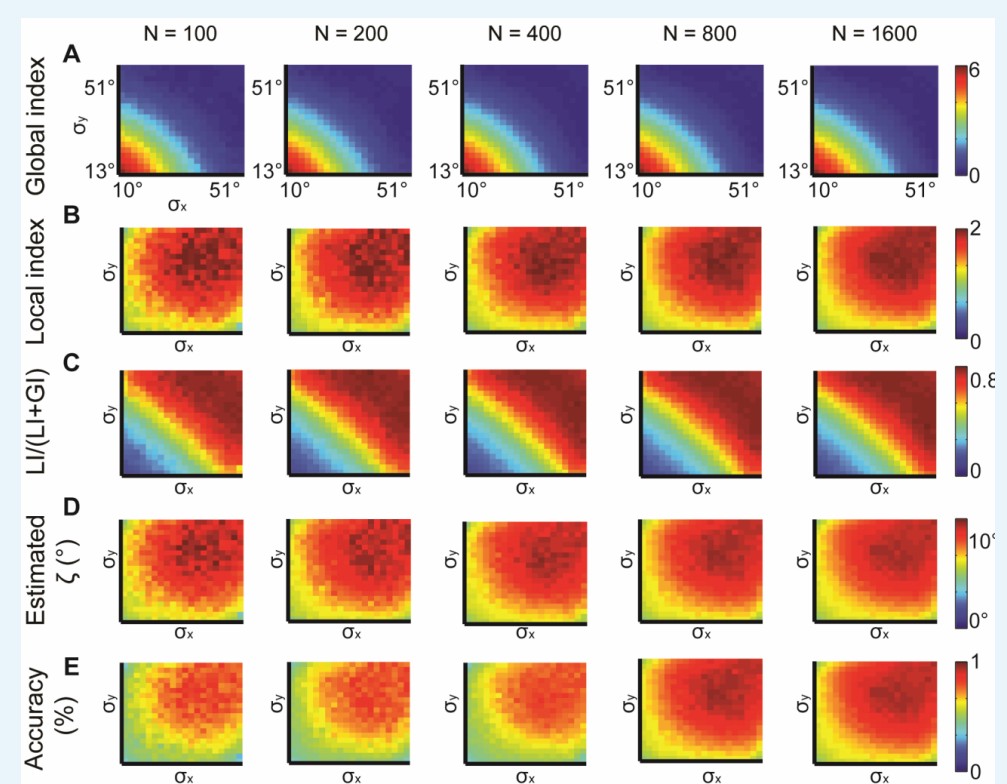

**Appendix 2—figure 3.** Simulations for different N, the number of observations. N = 100, 200, 400, 800, 1600 were examined. X, Y normal distributions with different $\sigma_x, \sigma_y$ and constant $\zeta = 15°$. All measures provide similar information but are noisier for lower N. (**A**) Global index. (**B**) Local index. (**C**) Relative contribution of LI to the observed alignment. (**D**) Retrieved estimated $\zeta$. (**E**) Accuracy of estimated $\zeta$.

## Appendix 3

### Simulating protein-protein co-localization

Co-localization of molecules is commonly used to predict potential local interactions under the assumption that the stoichiometry of interacting molecules remains constant across all observations. We demonstrated this by simulating a random variable $X$, representing molecular counts, and a random variable $Z$, representing the local interaction. Pairs of samples $x_i$ from $X$ and $\zeta_i$ from $Z$ were drawn, and $y_i = x_i\zeta_i$ represented the molecular counts of the co-localized molecule (**Appendix 3—figure 1A**). The joint distributions of $X, Y$ for four simulations are shown in **Appendix 3—figure 1B**. $X$ is normally distributed with mean $\mu = 0.5$ and standard deviations $\sigma = 0.2$, truncated to [0,1]. $Z$ is normally distributed with mean $\mu = 1$ and different standard deviations, simulating gradually increasing local interactions (**Appendix 3—figure 1B**, left-to-right). Applying DeBias on these data demonstrated that increased local interactions (reduced $\sigma_\zeta$) are translated to increased LI (**Appendix 3—figure 1C**).

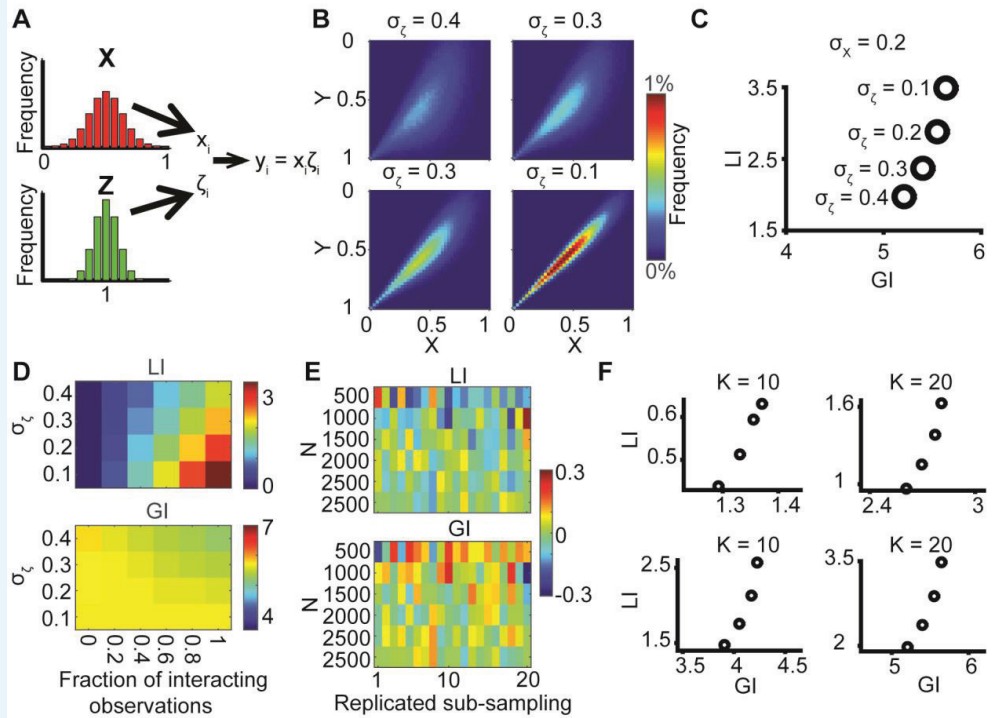

**Appendix 3—figure 1.** Simulating co-localization. (**A**) Simulation. We use the distributions $X = N(\mu_x = 0.5, \sigma_x), Z = N(\mu_\zeta = 1, \sigma_\zeta)$, where X is truncated to [0,1]. Pairs of coupled variables are constructed by drawing sample pairs $(x_i, \zeta_i)$ and constructing $y_i = x_i, \zeta_i$ (Materials and methods). (**B**) Simulated joint distributions. Shown are the joint distributions of 4 simulations with increased global bias (i.e., decreased $\mu_\zeta$). (**C**) Local and global indices calculated for the examples from panel B. Smaller $\zeta_i$ associate with larger LI. The weaker negative association between GI and $\sigma_\zeta$ is because larger $\sigma_\zeta$ induces a distribution Y that is more spread compared to X which reduces the GI. (**D**) Simulations of partial co-localization. A given fraction of observations for Y were calculated as shown in panels A–C, the rest were independently drawn from the distribution X implying no local interaction. Shown are LI (top) and GI (bottom) as functions of the fraction of locally interacting observations and $\sigma_\zeta$. LI associates with increased fraction of locally-interacting observations, whereas the effect is minor in GI, in accordance with panel C. (**E**) Deviation of LI, GI values reported in panel C as

functions of the number of observations n. 20 independent sub-sampling. Lower n associates with higher variability. No other trend is observed. (**F**) LI and GI patterns are independent of the number of alignment histogram bins K = 10–40. Equal size of dynamic ranges was set for LI and GI plots in panels C, D and F. K, number of histogram bins was set to 40 for all panels excluding F. $\sigma_x$ = 0.2 for panels B–E.

We next simulated a scenario, in which a partial subset of observations $y_i$ interacted with $x_i$; the remaining $y_i$ were drawn independently from the distribution $Y = X$. We found that LI values increased with the fraction of interacting observations (*Appendix 3—figure 1D*).

Finally, we assessed how variation in the quantization parameter (K) and number of observations (N) alter GI and LI (*Appendix 3—figure 1E-F*) and this was also demonstrated in our experimental data (*Figure 6—figure supplement 1G–J*).

Zaritsky *et al.* eLife 2017;6:e22323. DOI: 10.7554/eLife.22323

## Appendix 4

## Pixel- versus object-based co-localization

Many methods have been developed to quantify protein-protein co-localization. Pixel-based methods measure pixel-wise correlation coefficients (*Adler and Parmryd, 2010*; *Bolte and Cordelières, 2006*; *Costes et al., 2004*; *Manders et al., 1993*; *Pearson, 1901*), exploiting the notion that fluorescence levels of co-localized proteins are correlated. They suffer from background signal where no co-localization exists. Object-based methods first detect objects of interest and then assess co-localization based on second-order statistics of the spatial distributions of the detections (*Helmuth et al., 2010*; *Kalaidzidis et al., 2015*; *Lagache et al., 2015*; *Rizk et al., 2014*; *Semrau et al., 2011*). Object-based methods remove the background pixels but lose the information contained by the fluorescence levels at the detected objects. Thus, object-based methods are best applicable for the co-localization of binary signals (*Lagache et al., 2015*), but less suited for applications in which co-localization accounts for coupling of molecular counts on a continuous spectrum. Moreover, object-based methods require detection of objects in both channels, which often limits their applicability. Pros and cons for using either of these methods are presented in *Appendix 4—Table 1*. In the examples of receptor-CCP co-localization we implemented a hybrid of the two approaches: co-localization analysis by DeBias focused on the intensity of fluorescent readouts within detected CCPs to decompose the coupling of the two intensity variables into LI and GI (*Figure 6*). The same decomposition was demonstrated for a diffuse signal without objects in the PKC-FRET example (*Figure 5*).

**Appendix 4 – Table 1.** Object- versus pixel-based colocalization.

Object-based methods are best applicable for the colocalization of binary signals, but not/less for applications, in which colocalization accounts for coupling of molecular counts on a continuous spectrum.

| Pixel based | Object based |
| --- | --- |
| Pixel-wise correlation | Spatial colocalization |
| | Limited to object-detectable data |
| Suffer from background signal | |
| | Highly altered by detection accuracy |
| Affected by confounding factors (e.g., CCP size) | Lose the information at the detected objects |

