## [Decision Letter]

Thank you for submitting your article "Decoupling global biases and local interactions between cell biological variables" for consideration by *eLife*. Your article has been favorably evaluated by Naama Barkai (Senior Editor) and three reviewers, one of whom, Fabrice Cordelières (Reviewer #1) served as Reviewing Editor. The following individuals involved in review of your submission have agreed to reveal their identity: Chong Zhang (Reviewer #2); Perrine Paul-Gilloteaux (Reviewer #3).

The reviewers have discussed the reviews with one another and the Reviewing Editor has drafted this decision to help you prepare a revised submission.

Summary:

Zaritsky et al. propose a new approach to characterise interactions between two variables as a result of two influences: a global effect, independent of the two variables and influencing both, and a local effect that depends on their "direct" dependency. This paper raises the point of differentiating situations where correlations are observed, not due to a direct link between variables, but rather being the result of an overall behaviour. The authors built a framework in which both effects are reflected by two indicators: global and local indexes (GI and LI). The paper is divided into a theoretical part, where simulations are used to setup the framework, and a series of 4 biological examples where the "DeBias" method is applied.

Essential revisions:

1) The authors have performed simulation in which the bin widths of distributions or the number of observations have been progressively increased. Conclusions seem to be that a 15 quantization bins and a number of samples around 100 is acceptable. However, how should these parameters be set depending on the nature of the data to analyse? How should these parameters be set, depending on the distributions to analyse? Is there a generic rule? If so, this should be explained. If not, why did the author make those choices? The Methods section includes information about the number of quantization bins: why is it 89 for angular data, where as the value is 10 for PKC experiments and 39 for co-localization? From theoretical results, it seems LI or GI tends to (K-1)/2 when the other indicator is zero, K being the number of quantization bins. How would the authors recommend proceeding when the K parameter is not the same from one experiment to another? I fear we are missing proper guidelines.

2) For example of lack of usability: I did test it also with my own data of spot colocalization (simulation) and interestingly the GIs were inversely related to the spot density (but which was also related to the image size or N), and LIs was less obvious (my N was 100^2 and 256^2 with 100 spots each time with no colocalisation or with 50% of them colocalizing). I’ve found

GI no coloc 50% coloc

(squareroot size) 100 4.47 4.32

(squareroot size)256 3.67 3.62

LI no coloc 50% coloc

(squareroot size)100 1.59 2.24

(squareroot size)256 1.69 1.57

Before any conclusion, interpretation would require more characterisation work, or maybe provided with a simulation script in order to better apprehend the significance of it. I would have had to repeat the experiment, too laborious from the server, and in addition I do not know which number of bins was used in that case and if it was constant.

3) In terms of local interaction ζ, it is simulated as a ratio of θ by α, is this a generic assumption that local interaction is a variable linearly related to the alignment?

4) Subsection “Simulating synthetic data”, the authors demonstrated in the simulation the effectiveness of GI and LI. While it seems true by simulating local interaction of alignment by shifting one of the angles towards the other, how would GI and LI behave when the local interaction is not alignment, i.e., by shifting one away from the other? Would GI and LI show the opposite effect so as to be able to discriminate opposite local interactions?

5) Subsection “DeBias procedure”, for the adjustment #1 for the colocalization quantification, such normalization implicitly assumes the channel signals/intensities have linear relationship. This could be a strong assumption. What would be the adjustment for those signals that do not have linear relationship?

6) Probably some additional validation to highlight the method's advantage over other conventional intensity based colocalization methods would be useful. Such additional evaluation is probably not necessarily within the scope of a "Tools" submission, but that if such data were available, it would be a valuable scientific addition to the paper. Or at least some kind of generic guidelines about under which situations when DeBias may or may not be applied would be instructive.

7) The quantification power is actually relative: in the same conditions (meaning here same K, and potentially same N see point 2), the GIs and LIs would actually give relative information, but a twofold interaction factor would not show a twofold LI. In addition each experiment shows very different range of LIs and GIs. These values are theoretically in the range of [0; (k-1)/2], k been the number of bins, and does not denotes directly to the strength of the interaction, and have to be compared. Normalizing these descriptors by their upper bound may help to interpret.

8) In each of these experiments, the conclusions were drawn with a different path of reasoning and different constructions of statistical tests to compare and assess the significance of observed differences between these 2 descriptors for different conditions. I do believe that the manuscript would gain in impact by providing a general workflow (maybe a scheme with different possibilities), including the assessment of significance of the descriptors differences. Otherwise, the potential user of this powerful technique may struggle with the analysis and be led to erroneous conclusions.

9) The Discussion of this article is quite disappointing. The first part is a summary of previous conclusions, the second part being a short comparison to already published, approaching methods. The manuscript should benefit from this comparison being included into the proper section of the manuscript, namely the co-localisation one.

---

## [Author Response]

Essential revisions:

1) The authors have performed simulation in which the bin widths of distributions or the number of observations have been progressively increased. Conclusions seem to be that a 15 quantization bins and a number of samples around 100 is acceptable. However, how should these parameters be set depending on the nature of the data to analyse? How should these parameters be set, depending on the distributions to analyse? Is there a generic rule? If so, this should be explained. If not, why did the author make those choices? The Methods section includes information about the number of quantization bins: why is it 89 for angular data, where as the value is 10 for PKC experiments and 39 for co-localization? From theoretical results, it seems LI or GI tends to (K-1)/2 when the other indicator is zero, K being the number of quantization bins. How would the authors recommend proceeding when the K parameter is not the same from one experiment to another? I fear we are missing proper guidelines.

Setting K is a tradeoff between the number of observations and the range of observed values. In the manuscript, the narrow dynamic range in the PKC imaging and wider range in the CME co-localization experiments implied different selection of K.

In response to these valid questions by the reviewers, we have further validated that changing K does not alter the interpretation of simulated or experimental data analysis for both co-orientation and co-localization scenarios (Figure 3—figure supplement 1, Figure 6—figure supplement 1, Figure 10).

The Freedman-Diaconis rule is designed to calculate the histogram bin width so to minimize the difference between the area under the observed and the theoretical distribution (D Freedman, P Diaconis et al., 1981). Following the reviewers’ comments we implemented and integrated this rule to our source code and web-server implementation as a means to determine a default value of K and the marginal distributions’ bin width.

K must remain constant for a given application, for testing the alterations of LI/GI as response to different perturbations / experimental conditions. However, different applications may use different K values. This is emphasized in the Methods section (“Importantly, GI and LI across experiments can be compared only when evaluated with the same K value and this is enforced by the web-server.”).

*2) For example of lack of usability: I did test it also with my own data of spot colocalization (simulation) and interestingly the Gis was inversely related to the spot density (but which was also related to the image size or N), and Lis was less obvious ((my N was 100^2 and 256^2 with 100 spots each time with no colocalisation or with 50% of them colocalizing. I’ve found*

*GI no coloc 50%coloc*

*(squareroot size) 100 4.47 4.32*

*(squareroot size)256 3.67 3.62*

*LI no coloc 50%coloc*

*(squareroot size)100 1.59 2.24*

(squareroot size)256 1.69 1.57before any conclusion, interpretation would require more characterisation work, or maybe provided with a simulation script in order to better apprehend the significance of it. I would have had to repeat the experiment, too laborious from the server, and in addition I do not know which number of bins was used in that case and if it was constant.

It is not clear to us how these simulations were performed. Below we list possible explanation to the reviewer’s observations:

GI is inversely associated with the density of co-localized pixels. If the background pixels are drawn from a uniform distribution then more “background” pixels (in this case 6.5 fold) imply reduced GI, as observed.

LI is not increased when N = 256^2 (assuming 100 colocalized pixels). A fraction of ~0.0015 (100/255^2) co-localized pixels could be too subtle for DeBias to identify. In the case N = 100^2 and corresponding fraction of 1% co-localized pixels, this is easily identified (LI: 1.59 → 2.24).

In our new co-localization simulations we have mixed interacting and non-interacting observations and demonstrated that LI associates with an increased fraction of locally- interacting observations (Figure 10). In these simulations, the non- interacting observations from the 2^nd^ channel (Y) were drawn from the distribution of the first channel (X). Following the reviewer concern, we have also simulated very low fractions of interacting observations of 0%, 0.1%, 0.5% and 1% to highlight the sensitivity of DeBias. See Figure 11 (parameters were: σ_ζ_ = 0.1, σ_x_ = 0.2). We did not include these results in the manuscript because Figure 10 already captures a wide range of fractions of interacting observations. Its purpose was to demonstrate DeBias’ ability to identify scenarios of partial indications and this was achieved with the new figure panel.

Author response image 1.**DOI:**
http://dx.doi.org/10.7554/eLife.22323.018

Following the reviewers’ suggestion, we have made our co-localization and co-orientation simulation publicly available.

3) In terms of local interaction ζ, it is simulated as a ratio of θ by α, is this a generic assumption that local interaction is a variable linearly related to the alignment?

The calculation of LI and GI is based on the assumption that the local interaction is constant, implemented by modeling the LI as an additive constant by complementing GI (calculated from the marginal distributions) to the observed co-alignment / co-localization distribution. This assumption might be unrealistic in many biological systems. As DeBias is the first direct measure to quantify both local interactions and global bias, we simplified the estimation of LI and used it as a proxy to complex and more biological-realistic interactions. Our co-alignment simulations modeled the local interactions as a fraction of the alignment θ. We were able to show, even with this simplifying assumption, that we can discriminate and estimate ζ accurately (Figure 7). We have now also included a simulation for co-localization, where the local interactions are modeled as associations between the channels (y = αx). These results are shown in Appendix 3. Adjusting DeBias to non-constant bias is out-of-scope of this manuscript and will be pursued as future work.

4) Subsection “Simulating synthetic data”, the authors demonstrated in the simulation the effectiveness of GI and LI. While it seems true by simulating local interaction of alignment by shifting one of the angles towards the other, how would GI and LI behave when the local interaction is not alignment, i.e. by shifting one away from the other? Would GI and LI show the opposite effect so as to be able to discriminate opposite local interactions?

We have implemented simulations of two locally repulsive vector variables and demonstrated that yields negative LI values, indeed (Figure 2—figure supplement 1).

5) Subsection “DeBias procedure”, for the adjustment #1 for the colocalization quantification, such normalization implicitly assumes the channel signals/intensities have linear relationship. This could be a strong assumption. What would be the adjustment for those signals that do not have linear relationship?

This assumption was made to adjust the channels’ intensity to the [0,1] range. Indeed, this is a simplifying assumption that is now discussed in the new section on DeBias’ limitations in the Discussion. Adjusting the normalization to possibly non-linear interactions is out-of-scope of this manuscript.

6) Probably some additional validation to highlight the method's advantage over other conventional intensity based colocalization methods would be useful. Such additional evaluation is probably not necessarily within the scope of a "Tools" submission, but that if such data were available, it would be a valuable scientific addition to the paper. Or at least some kind of generic guidelines about under which situations when DeBias may or may not be applied would be instructive.

We already demonstrated that including GI increases discriminative abilities for the experimental CME co-localization data (Figure 6) and for simulated co-alignment data (Figure 2 – only subjectively). We now complement this by demonstrating enhanced discrimination performance of Pearson correlation by including the GI as a coupled measure (Figure 6—figure supplement 1). We also demonstrated the high association of LI and Pearson’s correlations (Pearson’s coefficient > 0.95), validating LI as a measure for local interactions.

7) The quantification power is actually relative: in the same conditions (meaning here same K, and potentially same N see point 2), the GIs and Lis would actually give relative information, but a twofold interaction factor would not show a twofold Li. In addition each experiment shows very different range of Lis and Gis. These values are theoretically in the range of [0; (k-1)/2], k been the number of bins, and does not denotes directly to the strength of the interaction, and have to be compared. Normalizing these descriptors by their upper bound may help to interpret.

It is true that the quantification power is relative. K can be set independently for every application to discriminate between different experimental conditions. It is also true that a two-fold increase in the direct interaction would not necessarily imply a two-fold increase in LI – this requires a priori knowledge of the nature of the interaction and adjustment of the calculation of LI. Tackling this issue for all of the four application areas seems outside the scope of this manuscript. When absolute values are needed, we will in the future refer to the DeBias framework and calibrate the LI change on an application-by- application basis. It should be noted that alternative measures for pixel-based co- localization are relative in nature as well. Normalizing by the upper bound will not solve the second concern (two-fold increased interaction mapped to a two-fold increase in LI). Since dividing by the constant K(K-1) will induce very low numbers for LI and GI (e.g., for K = 15 and LI = 1 → < 0.005) we have decided to avoid it in our implementation.

We excluded the possibility that N has a major effect on the calculated GI and LI values (point #2).

8) In each of these experiments, the conclusions were drawn with a different path of reasoning and different constructions of statistical tests to compare and assess the significance of observed differences between these 2 descriptors for different conditions. I do believe that the manuscript would gain in impact by providing a general workflow (maybe a scheme with different possibilities), including the assessment of significance of the descriptors differences. Otherwise, the potential user of this powerful technique may struggle with the analysis and be led to erroneous conclusions.

We included a general assessment of discrimination and integrated it into our source code and web-server implementation: the p-value of a two-sided Wilcoxon rank sum test for LI and GI of two distinct experimental conditions. In the web-server this part is now referred as ‘DeBias Analyst’.

In the manuscript, the only data assessed differently were the velocity-stress alignment measures (Figure 4) – because we had a single observation per condition and thus could not perform statistical analysis between independent experimental instances. This data was not produced in our lab (previously published data) so we could not retrieve more replicates and thus had to come up with a different analysis. Other methods were also proposed in the manuscript (e.g., classification accuracy, Figure 6) but are not provided in our code distribution.

9) The Discussion of this article is quite disappointing. The first part is a summary of previous conclusions, the second part being a short comparison to already published, approaching methods. The manuscript should benefit from this comparison being included into the proper section of the manuscript, namely the co-localisation one.

We feel that the summary part starting the Discussion is necessary because of the nature of the work. Showcasing DeBias on four quite distinct examples emphasizes the generalization of the method, especially, in extracting insightful mechanistic information that was not available in earlier co-localization quantification methods. This point is now emphasized in the ‘summary’ of the Discussion. Another reason for a summary is that many people read only the Discussion before they decide whether to invest time in reading the full text.

We did not follow the suggestion to move the comparison between pixel and object- based co-localization to the co-localization part of the manuscript. We feel that it will be confusing to follow a co-localization discussion in the Results section immediately before going into a general discussion of the features of DeBias analysis in the Discussion section. Having the comparison at the beginning of the co-localization results would interfere with the flow of the manuscript. Instead, we have rewritten the Discussion part addressing this issue and inserted a new Appendix 4 that discusses co-localization approaches in greater depth.

We also added a section on limitations of DeBias to the Discussion. It includes the issues raised by the reviewers: linearity assumption for co-localization normalization, adjusting DeBias to non-constant bias, relative quantification power, LI and GI are slightly coupled and that in some cases it takes additional means to interpret what biological properties are encoded in the GI.